# TFEB/Mitf links impaired nuclear import to autophagolysosomal dysfunction in C9-ALS

Kathleen M Cunningham[1], Kirstin Maulding[1], Kai Ruan[2], Mumine Senturk[3], Jonathan C Grima[4,5], Hyun Sung[2], Zhongyuan Zuo[6], Helen Song[2], Junli Gao[7], Sandeep Dubey[2], Jeffrey D Rothstein[1,2,4,5], Ke Zhang[7], Hugo J Bellen[3,6,8,9,10], Thomas E Lloyd[1,2,5]*

[1]Cellular and Molecular Medicine Program, School of Medicine, Johns Hopkins University, Baltimore, United States; [2]Department of Neurology, School of Medicine, Johns Hopkins University, Baltimore, United States; [3]Program in Developmental Biology, Baylor College of Medicine (BCM), Houston, United States; [4]Brain Science Institute, School of Medicine, Johns Hopkins University, Baltimore, United States; [5]Solomon H. Snyder Department of Neuroscience, School of Medicine, Johns Hopkins University, Baltimore, United States; [6]Department of Molecular and Human Genetics, BCM, Houston, United States; [7]Department of Neuroscience, Mayo Clinic, Jacksonville, United States; [8]Department of Neuroscience, BCM, Houston, United States; [9]Jan and Dan Duncan Neurological Research Institute, Texas Children's Hospital, Houston, United States; [10]Howard Hughes Medical Institute, Houston, United States

**Abstract** Disrupted nucleocytoplasmic transport (NCT) has been implicated in neurodegenerative disease pathogenesis; however, the mechanisms by which disrupted NCT causes neurodegeneration remain unclear. In a *Drosophila* screen, we identified *ref(2)P/p62*, a key regulator of autophagy, as a potent suppressor of neurodegeneration caused by the GGGGCC hexanucleotide repeat expansion (G4C2 HRE) in *C9orf72* that causes amyotrophic lateral sclerosis (ALS) and frontotemporal dementia (FTD). We found that p62 is increased and forms ubiquitinated aggregates due to decreased autophagic cargo degradation. Immunofluorescence and electron microscopy of *Drosophila* tissues demonstrate an accumulation of lysosome-like organelles that precedes neurodegeneration. These phenotypes are partially caused by cytoplasmic mislocalization of Mitf/TFEB, a key transcriptional regulator of autophagolysosomal function. Additionally, TFEB is mislocalized and downregulated in human cells expressing GGGGCC repeats and in C9-ALS patient motor cortex. Our data suggest that the *C9orf72*-HRE impairs Mitf/TFEB nuclear import, thereby disrupting autophagy and exacerbating proteostasis defects in C9-ALS/FTD.

*For correspondence:
tlloyd4@jhmi.edu

## Introduction

A GGGGCC (G4C2) hexanucleotide repeat expansion (HRE) in chromosome nine open reading frame 72 (*C9orf72*) is the most common genetic cause of amyotrophic lateral sclerosis (ALS) and frontotemporal dementia (FTD), accounting for up to 40% of cases of familial ALS (*DeJesus-Hernandez et al., 2011*; *ITALSGEN Consortium et al., 2011*). ALS and/or FTD caused by mutations in *C9orf72* (C9-ALS/FTD) is inherited in an autosomal dominant manner, suggesting that the HRE causes disease through gain-of-function or haploinsufficiency (*DeJesus-Hernandez et al., 2011*; *Ling et al., 2013*). Loss of C9orf72 function has been linked to disruption of autophagy and

lysosome function, though neurodegeneration is not observed in *C9orf72* knockout mice (*Liu et al., 2016*; *Shi et al., 2018*; *Webster et al., 2016*), suggesting that C9-ALS/FTD is primarily caused by toxicity of the HRE. Furthermore, expression of G4C2 repeats causes neurotoxicity in *Drosophila* and cell culture models of C9-ALS (*Goodman et al., 2019a*; *Kramer et al., 2016*; *Tran et al., 2015*). This toxicity has been proposed to occur through either G4C2 repeat RNA-mediated sequestration of RNA-binding proteins or translation of the G4C2 repeats into dipeptide-repeat proteins (DPRs) through non-canonical repeat-associated non-AUG (RAN) translation (*Donnelly et al., 2013*; *Goodman et al., 2019a*; *Mori et al., 2013*; *Tran et al., 2015*).

We previously conducted a *Drosophila* screen of candidate proteins that bound with moderate-to-high affinity to G4C2 RNA and identified modulation of the nucleocytoplasmic transport (NCT) pathway as a potent modifier of G4C2 toxicity in both fly and iPS neuron models of C9-ALS (*Zhang et al., 2015a*), a finding that has also been made by other groups (*Freibaum et al., 2015*; *Jovičić et al., 2015*). The mechanisms by which the G4C2 HRE disrupts NCT remain unclear, but potential mechanisms include G4C2 RNA binding to the master NCT regulator RanGAP (*Zhang et al., 2015a*), DPRs disrupting the nuclear pore complex (*Boeynaems et al., 2016*; *Shi et al., 2017*; *Zhang et al., 2016*), stress granules sequestering NCT factors (*Zhang et al., 2018*), or cytoplasmic TDP-43-dependent dysregulation of karyopherin-α (*Chou et al., 2018*; *Gasset-Rosa et al., 2019*; *Solomon et al., 2018*). Recently, a role for NCT disruption in Huntington's disease and Alzheimer's disease has been proposed, indicating that NCT disruption may be a common mechanism in several neurodegenerative diseases (*Eftekharzadeh et al., 2018*; *Gasset-Rosa et al., 2017*; *Grima et al., 2017*). However, the pathways affected by NCT disruption that cause neurodegeneration have not yet been elucidated.

In a *Drosophila* screen for modifiers of G4C2-mediated neurodegeneration (*Zhang et al., 2015a*), we identified refractory to sigma P (*ref(2)P*), the *Drosophila* homolog of *p62/SQSTM1* (Sequestosome 1). *p62/SQSTM1* functions in macroautophagy (hereafter termed autophagy), and mutations in *p62/SQSTM1* are a rare genetic cause of ALS/FTD (*Cirulli et al., 2015*; *Le Ber et al., 2013*; *Teyssou et al., 2013*). Interestingly, many other genes implicated in ALS/FTD function in autophagy (*Evans and Holzbaur, 2019*; *Lin et al., 2017*; *Ramesh and Pandey, 2017*) such as tank-binding kinase 1 (*TBK1*), optineurin (*OPTN1*), ubiquilin 2 and 4 (*UBQLN2 and 4*), valosin-containing protein (*VCP*), charged multivesicular body protein 2B (*CHMP2B*), VAMP-associated protein B (*VapB*), and the C9orf72 protein itself (*O'Rourke et al., 2015*; *Sellier et al., 2016*; *Sullivan et al., 2016*; *Ugolino et al., 2016*; *Webster et al., 2016*; *Yang et al., 2016*). Organelles and protein aggregates are degraded via polyubiquitination and targeting to a newly forming autophagosome, followed by degradation upon fusion with the lysosome. Deletion of key autophagy genes in neurons is sufficient to cause neurodegeneration in mice (*Hara et al., 2006*; *Komatsu et al., 2006*).

Although autophagy and nucleocytoplasmic transport have both been implicated in neurodegeneration, it is unclear whether or how these two pathways interact in disease pathogenesis (*Gao et al., 2017*; *Thomas et al., 2013*). Here, we show that expression of expanded G4C2 repeats is sufficient to disrupt autophagy in *Drosophila*, leading to an accumulation of p62 and ubiquitinated protein aggregates. We find that autophagolysosomal defects are caused by loss of nuclear localization of the transcription factor Mitf (the *Drosophila* homolog of TFEB), which regulates transcription of genes involved in autophagolysosome biogenesis (*Bouché et al., 2016*; *Palmieri et al., 2011*; *Sardiello et al., 2009*; *Zhang et al., 2015b*). Furthermore, suppressing this NCT defect is sufficient to rescue Mitf nuclear localization, restoring autophagy and lysosome function and rescuing neurodegeneration. These findings suggest a pathogenic cascade in C9-ALS/FTD whereby NCT disruption causes a failure of autophagosome biogenesis and lysosome dysfunction that ultimately leads to neuronal death.

## Results

### Ref(2)P/p62 knockdown suppresses G4C2-mediated neurodegeneration

Expression of 30 G4C2 repeats (*30R*) in the eye using *GMR-Gal4* results in progressive photoreceptor degeneration and visible ommatidial disruption by day 15 (*Figure 1A*; *Xu et al., 2013*; *Zhang et al., 2015a*). In a genetic modifier screen of over 800 RNAi lines, *UAS-ref(2)P^RNAi* was among the strongest of 32 suppressors of G4C2-mediated eye degeneration (*Zhang et al.,*

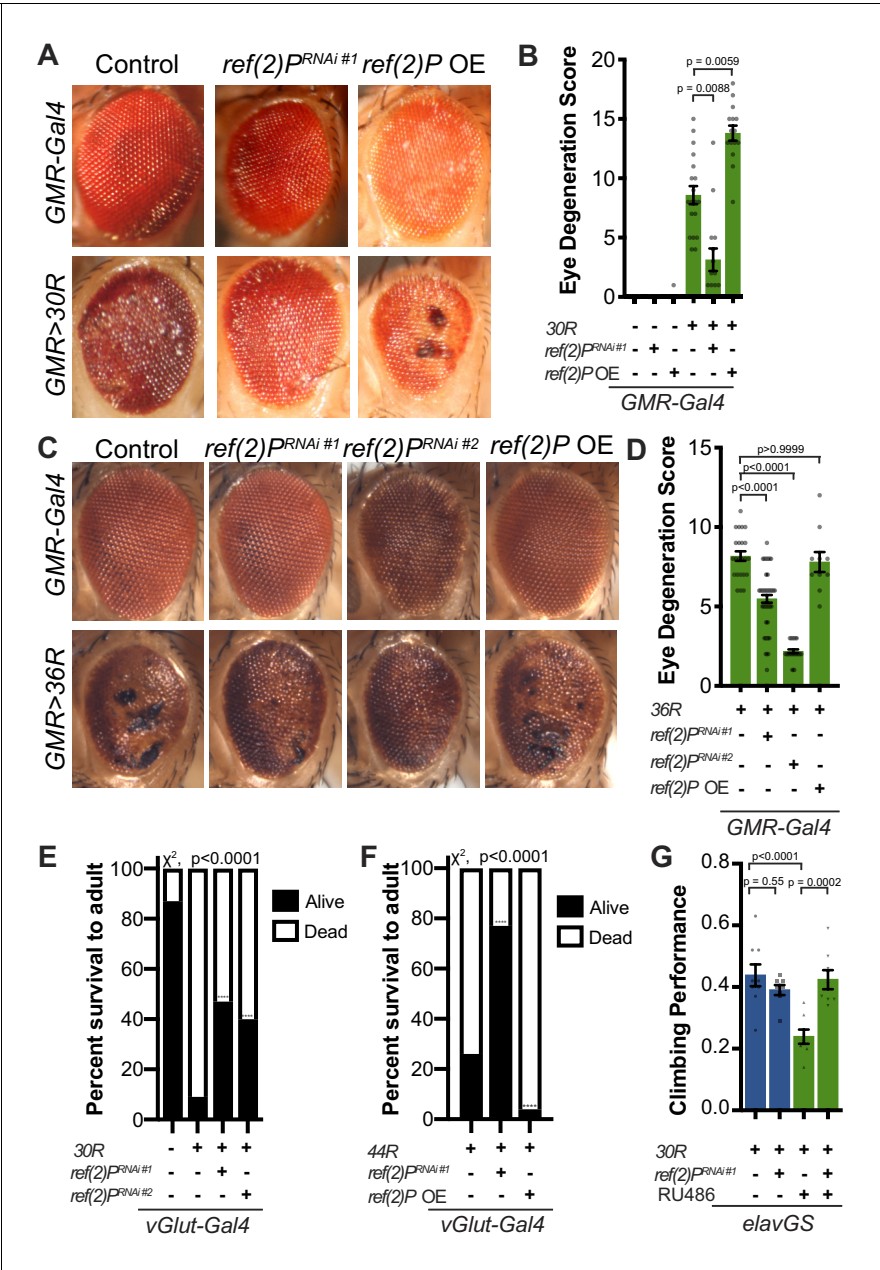

**Figure 1.** Autophagy receptor Ref(2) P/p62 genetically suppresses G4C2-HRE-mediated degeneration. (A) 15-day-old *Drosophila* eyes expressing *GMR-Gal4* +/- *UAS-30R* (*GMR>30R*) with RNAi background (control), *ref(2)P* $^{RNAi\#1}$ or overexpression (OE) of *ref(2)P*. (B) Quantification of external eye degeneration in A by semi-quantitative scoring system. Data are reported as mean ± SEM. Kruskal-Wallis test, p<0.0001, followed by Dunn's multiple comparisons, n ≥ 15 adults. (C) 15-day-old *Drosophila* eyes expressing *GMR-Gal4* +/- *UAS-36R* (*GMR >36R*) along with *UAS-luciferase* $^{RNAi}$ (control), *UAS-ref(2)P* $^{RNAi\ \#1}$, *UAS-ref(2)P* $^{RNAi\ \#2}$, or *UAS-ref(2)P* OE. (D) Quantification of external eye degeneration in C by semi-quantitative scoring system. Data are reported as mean ± SEM. Kruskal-Wallis test, p<0.0001, followed by Dunn's multiple comparisons, n = 23, 62, 28, 10 adults respectively. (E) Percent of pupal eclosion of adult flies expressing the motor neuron driver *vGlut-Gal4* +/- *UAS-30R* and RNAi background control or *UAS-ref(2)P* $^{RNAi\ \#1}$. Fisher's exact test, n ≥ 100 pupa. (F) Percent of pupal eclosion of adult flies expressing the motor neuron driver *vGlut-Gal4* +/- *UAS-44R* along with *UAS-luciferase RNAi*, *UAS-ref(2)P* $^{RNAi\ \#1}$, or *UAS-ref(2)P* OE. Fisher's exact test, n ≥ 55 pupa. (G) Adult *Drosophila* expressing *UAS-30R* under the control of the inducible, pan-neuronal *elavGS* induced with 200 μM RU486 or vehicle alone and co-expressing control or *UAS-ref(2)P* $^{RNAi\ \#1}$. Data are reported as mean ± SEM. One-way ANOVA, ****p<0.0001, with Sidak's multiple comparisons test, n = 9, 8, 8, 8 groups of 10 flies.
The online version of this article includes the following figure supplement(s) for figure 1:

**Figure supplement 1.** Ref(2)P/p62 genetically modifies G4C2-HRE.

*2015a*; *Figure 1A*). *ref(2)P* is the *Drosophila* homolog of *p62/SQSTM1*, and this modifier is of particular interest because *SQSTM1* mutations that cause loss of selective autophagy cause ALS/FTD (*Cirulli et al., 2015*; *Goode et al., 2016*; *Le Ber et al., 2013*), and p62 aggregates are pathological features of both familial and sporadic ALS (*Al-Sarraj et al., 2011*; *Cooper-Knock et al., 2012*). Knockdown of *ref(2)P* suppresses eye degeneration, whereas overexpression of *ref(2)P* enhances this phenotype (*Figure 1A–B*, *Figure 1—figure supplement 1A*). *ref(2)P* [RNAi #1] expression reduced *ref (2)P* mRNA levels by ~80%, but did not alter G4C2 RNA levels in *30R* expressing eyes (*Figure 1—figure supplement 1B–C*), suggesting that *ref(2)P* acts downstream of G4C2 transcription. Similarly, knockdown of *ref(2)P* also rescued eye degeneration in a second G4C2 model expressing 36 G4C2 repeats (*36R*) (*Mizielinska et al., 2014*; *Figure 1C–D*). We next assessed the ability of *ref(2)P* [RNAi] to rescue toxicity of G4C2 repeats in motor neurons using the *30R* model and a new G4C2 model expressing *44R* (*Goodman et al., 2019b*). As shown in *Figure 1E–F*, while expression of either *30R* or *44R* in motor neurons with *vGlut-Gal4* leads to paralysis and lethality during pupal development, knockdown of *ref(2)P* partially rescues this phenotype, whereas overexpression of *ref(2)P* enhances the pupal lethality observed with *44R* expression. These data suggest that *ref(2)P* is required for G4C2-mediated toxicity during *Drosophila* development.

To determine whether *ref(2)P* knockdown is able to suppress age-dependent neurodegeneration, we used a pan-neuronal, inducible 'GeneSwitch' driver (*elavGS*) in which *30R*-expression leads to a marked reduction in climbing ability after 7 days (*Figure 1G*). This climbing defect is suppressed with coexpression of *ref(2)P* [RNAi], suggesting that *ref(2)P* contributes to G4C2-mediated neurotoxicity in the adult nervous system. Since RAN-translation of arginine-containing DPRs have been implicated in G4C2-mediated toxicity in *Drosophila* (*Kwon et al., 2014*; *Mizielinska et al., 2014*), we next tested whether *ref(2)P* knockdown rescues poly-glycine-arginine (GR) repeat-mediated toxicity. As shown in *Figure 1—figure supplement 1D*, *ref(2)P* [RNAi] partially rescues the severe eye degeneration phenotype caused by poly(GR)$_{36}$ expression. Together, these data indicate that *ref(2)P*, the *Drosophila* orthologue of *p62/SQSTM1*, modulates G4C2-mediated neurodegeneration.

## G4C2 repeat expression impairs autophagic flux

p62/SQSTM1-positive inclusions are a common pathologic feature seen in brains of C9-ALS/FTD patients where they colocalize with ubiquitin and DPRs (*Al-Sarraj et al., 2011*). We next investigated the localization of Ref(2)P protein (hereafter referred to as p62) in motor neurons. Expression of *30R* leads to the formation of many large p62:GFP puncta in cell bodies compared to controls that strongly colocalize with poly-ubiquitinated proteins (*Figure 2A–B*, *Figure 2—figure supplement 1A–B*). Western blot analysis demonstrates that p62 and poly-ubiquitin are strongly upregulated in flies ubiquitously expressing *30R* (*Figure 2C*, *Figure 2—figure supplement 1C–D*). Similarly, immunofluorescence staining with a p62 antibody shows endogenous p62 accumulations colocalizing with polyubiquitinated proteins in the ventral nerve cord and salivary gland of flies ubiquitously overexpressing *30R* (*Figure 2—figure supplement 1E*). These data show that G4C2 repeat expression in fly models recapitulates the p62 accumulation with ubiquitinated protein aggregates seen in C9-ALS/FTD patient tissue and iPS neurons (*Almeida et al., 2013*; *Mackenzie et al., 2014*).

Increased p62 levels can be due to either increased transcription and/or translation or insufficient protein degradation (*Korolchuk et al., 2010*). Using qRT-PCR, we find that *ref(2)P* transcript levels are unchanged in G4C2 repeat-expressing larvae (*Figure 2—figure supplement 1F*), suggesting that G4C2 repeats cause p62 upregulation by inhibiting p62 degradation. Since p62 is degraded by autophagy and disrupted autophagic flux is known to cause p62 upregulation, we assessed autophagy in G4C2-repeat-expressing flies. We first co-expressed the tagged autophagosome marker mCherry:Autophagy-related 8 (Atg8, the fly orthologue of mammalian Microtubule-associated protein 1A/1B-light chain 3 (LC3)) with *30R* in fly motor neurons and found a marked reduction in mCherry:Atg8 autophagic esicles (AVs) when compared to wild-type controls (*Figure 2D–E*). p62: GFP accumulation and loss of mCherry:Atg8 puncta were recapitulated in *36R* and poly(GR)$_{36}$ *Drosophila* models of C9-ALS/FTD (*Figure 2—figure supplement 1G–J*). Reduction of mCherry:Atg8-positive vesicles coupled with the accumulation of p62 and ubiquitin suggest that autophagic flux is impaired in these fly models of C9-ALS/FTD.

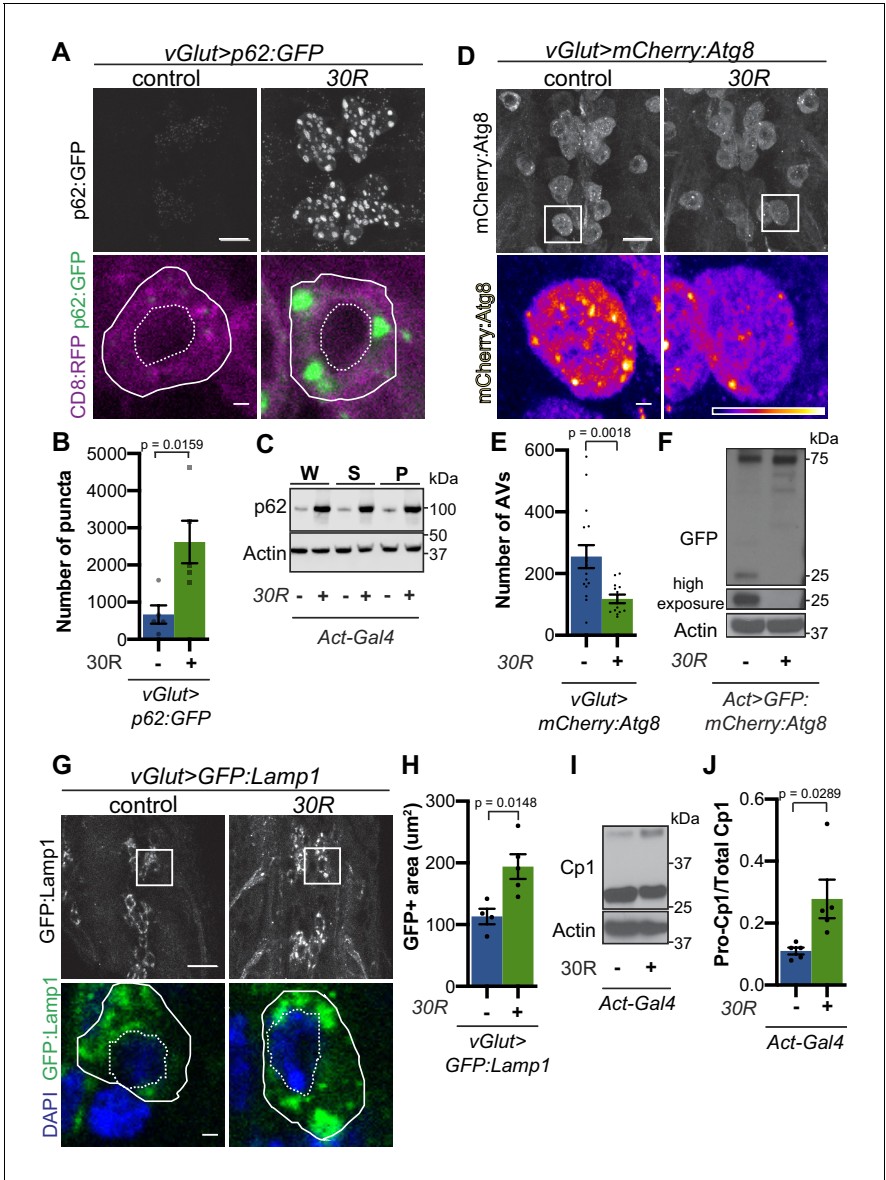

**Figure 2.** G4C2 repeat expression impairs autophagic flux. (**A**) *Drosophila* motor neurons expressing *UAS-p62:GFP +/- UAS-30R*, showing multiple motor neuron cell bodies (top) or a representative cell co-expressing the membrane marker CD8:RFP (bottom). Plasma membrane outlined with solid white line; nucleus outlined with dotted line. Scale bar = 10 μm (top), 1 μm (bottom) (**B**) Quantification of number of p62:GFP puncta in *Drosophila* motor neuron cell bodies. Data are reported as mean ± SEM. Mann-Whitney test, n = 5 larvae per genotype. (**C**) Western blot of anti-p62 and anti-beta-actin showing the whole (W), supernatant (S) and pellet (P) fractions of lysates from *Drosophila* larvae ubiquitously expressing -/+ *UAS-30R* under the control of *Act-Gal4*. (**D**) *Drosophila* motor neurons expressing *UAS-mCherry:Atg8 -/+ UAS-30R* showing cell bodies (top) with an example single cell highlighting mCherry:Atg8-positive puncta (bottom). Scale bar = 10 μm (top), 1 μm (bottom). (**E**) Quantification of mCherry:Atg8-positive autophagic vesicles (AVs) in the ventral nerve cord of *vGlut-Gal4/+* or *vGlut >30R* expressing flies. Data are reported as mean ± SEM. Mann-Whitney test, n = 16 and 13 larvae, respectively. (**F**) Western blot of anti-GFP and anti-beta-actin of lysates from whole *Drosophila* larvae ubiquitously expressing *UAS-GFP: mCherry:Atg8 -/+ UAS-30R* under the control of *Act-Gal4* showing full length GFP:mCherry:Atg8 at 75 kDa and cleaved GFP at 25 kDa. (**G**) *Drosophila* motor neurons expressing *UAS-GFP:Lamp1* (with N-terminal [luminal] GFP) *-/+ UAS-30R* under the control of *vGlut-Gal4* in multiple cell bodies (top) or in a representative cell (bottom). Scale bar = 10 μm (top), 1 μm (bottom). (**H**) Quantification of GFP:Lamp1 positive area in G. Data are reported as mean ± SEM. Student's t-test, n = 5 larvae. (**I**) Western of whole *Act-Gal4 Drosophila* larvae -/+ *UAS-30R* blotted for the lysosomal protease Cp1, showing pro- (inactive, upper band) and cleaved (active, lower band) Cp1. (**J**) Quantification of the ratio of pro-Cp1 to total Cp1 in I. Data are reported as mean ± SEM. Student's t-test, n = 5 biological replicates.

The online version of this article includes the following figure supplement(s) for figure 2:

**Figure supplement 1.** p62:GFP accumulates in C9-ALS fly models and co-localizes with poly-ubiquitin.

**Figure supplement 2.** Rescuing G4C2-mediated lysosome defects reduces neurodegeneration.

## G4C2 repeat expression causes lysosome defects

To further study lysosomal morphology and function, we expressed Lysosome- associated membrane protein 1 (Lamp1) with luminally-tagged GFP in our control and G4C2-expressing flies. Since GFP is largely quenched by the acidity of lysosomes in control animals (*Pulipparacharuvil et al., 2005*), the accumulation of GFP:Lamp-positive vesicles in *30R*-expressing motor neurons suggests a defect in lysosomal acidity or targeting of GFP:Lamp to mature lysosomes (*Figure 2G–H*). Furthermore, we observe a marked increase in size and number of late endosomes and lysosomes using genomically tagged Ras-related GTP-binding protein 7, Rab7:YFP, throughout *30R*-expressing motor neurons (*Figure 2—figure supplement 2A*) without alterations in early endosomes labeled with Rab5:YFP (data not shown). Together, these data demonstrate a marked expansion of the late endosome/lysosome compartment in G4C2-expressing neurons.

Though accumulation of p62 and ubiquitinated proteins could be caused by a failure of autophagic vesicles to fuse with the degradative endolysosomal compartment, we did not detect a decrease in mCherry:Atg8+, Rab7:GFP+ amphisomes in G4C2-expressing motor neuron cell bodies (*Figure 2—figure supplement 2B–F*). To assess autophagolysosomal function after fusion, we performed a GFP liberation assay on larvae expressing GFP:mCherry:Atg8 (*Klionsky et al., 2016*;

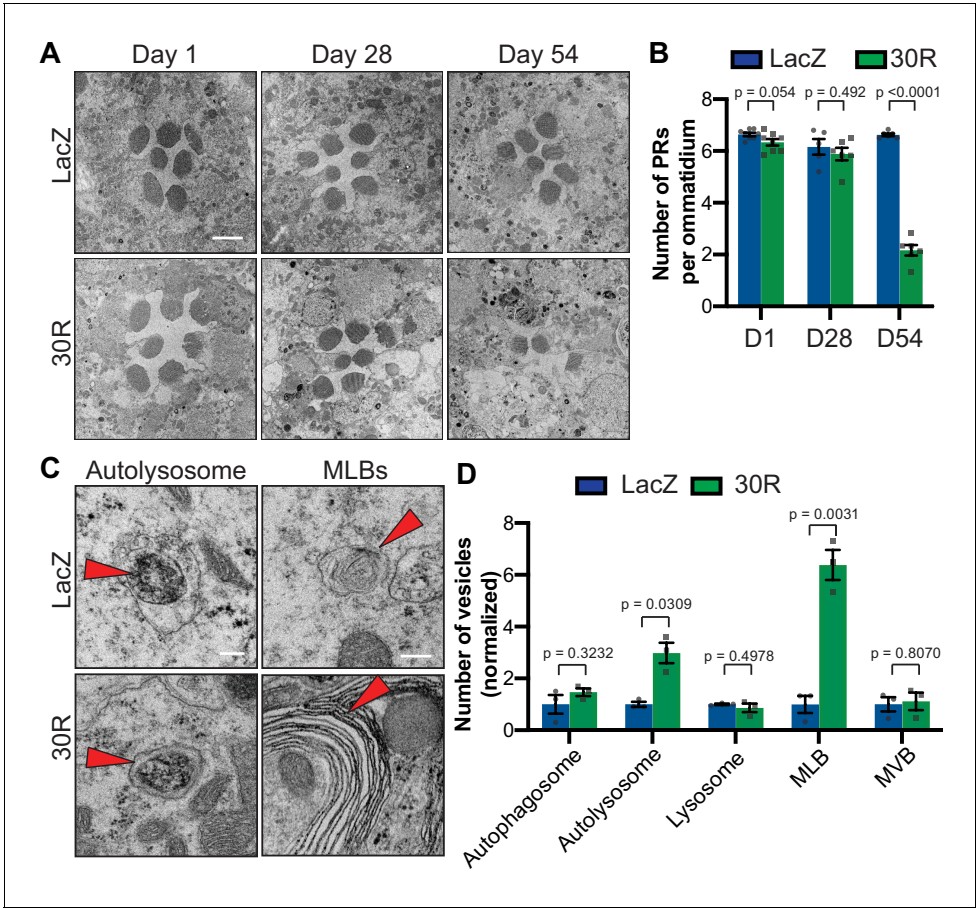

**Figure 3.** Autophagolysosomal defects precede neurodegeneration in photoreceptor neurons. (**A**) Transmission electron microscopy (TEM) of rhabdomeres (cell bodies) in *Rhodopsin1-Gal4* (*Rh1-Gal4*) driving *UAS-LacZ* (control) or *UAS-30R* at Day 1, Day 28, and Day 54 after eclosion. Scale bar = 2 μm. (**B**) Quantification of number of healthy (not split) photoreceptors (PRs) per ommatidium in A. Data are reported as mean ± SEM. Student's t-test, n = 8, 8, 6, 6, 6, and 6 flies, respectively. (**C**) TEM images at 28 days of *Drosophila* eyes (rhabdomeres) -/+ 30R repeats expressed by *Rh1-Gal4* showing representative autolysosomes and multilamellar bodies (MLBs), marked with red arrows. Scale bar = 200 nm. (**D**) Quantification of different vesicle types (autophagosomes, autolysosomes, lysosomes, MLBs, and multivesicular bodies (MVBs)) shown in TEM of rhabdomeres with *Rh1-Gal4* driving *UAS-LacZ* or *UAS-30R* (as in C) normalized to LacZ (control). Data are reported as mean ± SEM. Student's t-test, n = 3 adults per genotype. The online version of this article includes the following figure supplement(s) for figure 3:

**Figure supplement 1.** Progressive synapse degeneration in G4C2-expressing photoreceptor neurons.

*Mauvezin et al., 2014*). GFP is degraded more slowly than the rest of the mCherry:Atg8 protein, leaving a population of free GFP in functioning lysosomes. Free lysosomal GFP is not observed in G4C2-expressing larvae, suggesting an impairment in GFP:mCherry:Atg8 degradation by the lysosome (*Figure 2F*). To directly probe lysosome enzymatic activity, we performed Western analysis of *Drosophila* cathepsin Cp1. Whereas pro-Cp1 is normally cleaved to its mature form by acid hydrolases in lysosomes (*Kinser and Dolph, 2012*), larvae ubiquitously expressing *30R* show an increase in the ratio of pro-Cp1 to Cp1, indicating a decrease in pro-Cp1 cleavage efficiency (*Figure 2I–J*). Together, these data suggest that lysosomes are expanded and dysfunctional in G4C2 repeat-expressing animals.

To investigate whether the autophagic pathway defects precede neurodegeneration in G4C2 repeat-expressing neurons, we performed transmission electron microscopy (TEM) on *Drosophila* eyes. As *GMR-Gal4* is expressed throughout the development of the eye, we chose to perform electroretinograms (ERGs) of fly eyes selectively expressing *30R* in photoreceptor neurons (PRs) using *Rh1-Gal4*, which turns on during adulthood. *Rh1 >30R* PRs show only a mild reduction of ON transient amplitude at 28 days, but a complete loss of ON and OFF transients and a decrease in ERG amplitude by 56 days (*Figure 3—figure supplement 1A–D*), indicating a slow and progressive loss of synaptic transmission and impaired phototransduction respectively. These changes also correspond to a marked loss of photoreceptors and synaptic terminals by 54 days which are not observed at 28 days (*Figure 3A–B*; *Figure 3—figure supplement 1E*). We therefore examined autophagic structures by TEM at 28 days, prior to cell loss. Strikingly, we observe a marked increase in the size and number of multilamellar bodies (MLBs) (*Figure 3C–D*). MLBs are commonly observed in lysosomal storage diseases and result from a deficiency of lysosomal hydrolases and accumulations of lysosomal lipids and membranes (*Hariri et al., 2000*; *Weaver et al., 2002*). Though we did not detect an alteration in the number of autophagosomes, lysosomes, or multivesicular bodies, we did see a significant increase in the number of autolysosomes (*Figure 3C–D*). These data suggest that autophagolysosomal function is disrupted in G4C2-expressing photoreceptor neurons at early stages of degeneration.

Given the impairment in autophagic flux, we hypothesized that genetic or pharmacologic manipulations that accelerate autophagy may suppress neurodegenerative phenotypes, whereas those that further impede autophagy would enhance the phenotypes. Indeed, in a candidate-based screen, activation of early steps in the autophagic pathway (e.g. by *Atg1* overexpression) suppresses eye degeneration and blocking autophagosome/lysosome fusion (e.g. by *Snap29* knockdown) enhances eye degeneration (*Supplementary file 1*). Similarly, pharmacologic activation of autophagy via inhibition of mTor with rapamycin or mTor-independent activation via trehalose (*Sarkar et al., 2007*) rescues neurodegenerative phenotypes and p62 accumulation (*Figure 2—figure supplement 2G–K*). Together, these data show that promoting autophagy or lysosomal fusion are potent suppressors of G4C2-mediated neurodegeneration.

## Nucleocytoplasmic transport impairment disrupts autophagic flux

A diverse array of cellular pathways including autophagy, RNA homeostasis, and NCT are implicated in the pathogenesis of ALS and FTD (*Balendra and Isaacs, 2018*; *Evans and Holzbaur, 2019*; *Gao et al., 2017*; *Lin et al., 2017*; *Ling et al., 2013*; *Ramesh and Pandey, 2017*). However, the sequence of events in the pathogenic cascade remains unknown. Cytoplasmic protein aggregates or RNA stress granule formation is sufficient to disrupt nucleocytoplasmic transport (*Woerner et al., 2016*; *Zhang et al., 2018*). We therefore tested whether defects in autophagy are upstream, downstream, or in parallel with defects in NCT.

We first tested whether knockdown of *ref(2)P* rescues the mislocalization of the NCT reporter shuttle-GFP (S-GFP) containing both a nuclear localization sequence (NLS) and nuclear export sequence (NES). G4C2 repeat expression causes mislocalization of S-GFP to the cytoplasm (*Zhang et al., 2015a*), but knockdown of *ref(2)P* does not restore nuclear localization (*Figure 5—figure supplement 1A*). Similarly, stimulation of autophagy with rapamycin or trehalose fails to rescue S-GFP mislocalization in G4C2 expressing salivary glands (*Figure 5—figure supplement 1B*). Stimulating autophagy does not rescue NCT defects although it can rescue neurodegeneration, suggesting that autophagy defects are either independent of or downstream of NCT defects. Indeed, *RanGAP* knockdown increases the number and size of p62:GFP puncta, similar to the effects of

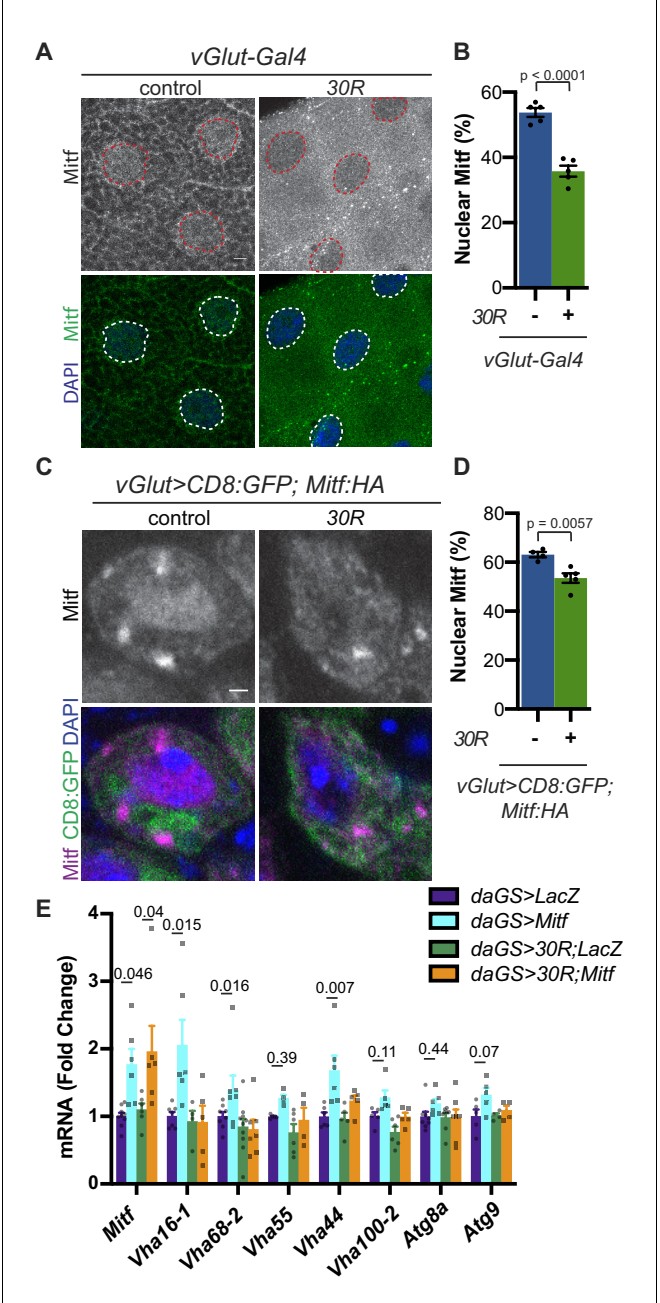

**Figure 4.** Mitf/TFEB is mislocalized from the nucleus and inactivated. (**A**) *Drosophila* larval salivary glands -/+ *UAS-30R* under the control of *vGlut-Gal4* stained with anti-Mitf and DAPI. Dotted lines outline nuclei. Scale bar = 10 µm. (**B**) Quantification of percent (%) nuclear Mitf (nuclear Mitf fluorescence/total fluorescence) in A. Data are reported as mean ± SEM. Student's t-test, n = 5 larvae per genotype. (**C**) *Drosophila* motor neurons (MNs) expressing *UAS-Mitf-HA and UAS-CD8:GFP* -/+ *UAS-30R* under the control of *vGlut-Gal4* stained with anti-HA, anti-GFP (membrane), and DAPI to show nuclear localization. Scale bar = 1 µm. (**D**) Quantification of percent (%) nuclear Mitf in C. Data are reported as mean ± SEM. Student's t-test, n = 4 and 5 larvae, respectively, with at least 10 motor neurons per larva. (**E**) Quantitative RT-PCR to assess transcript levels of *Mitf* and seven target genes from lysates of *Drosophila* heads expressing control (*UAS-LacZ*) or *UAS-30R* driven by *daGS* in control conditions or with overexpression of *Mitf*. Data are reported as mean ± SEM. One-way ANOVA, p<0.0001, with Sidak's multiple comparisons test, n ≥ 4 biological replicates of 30 heads per genotype.

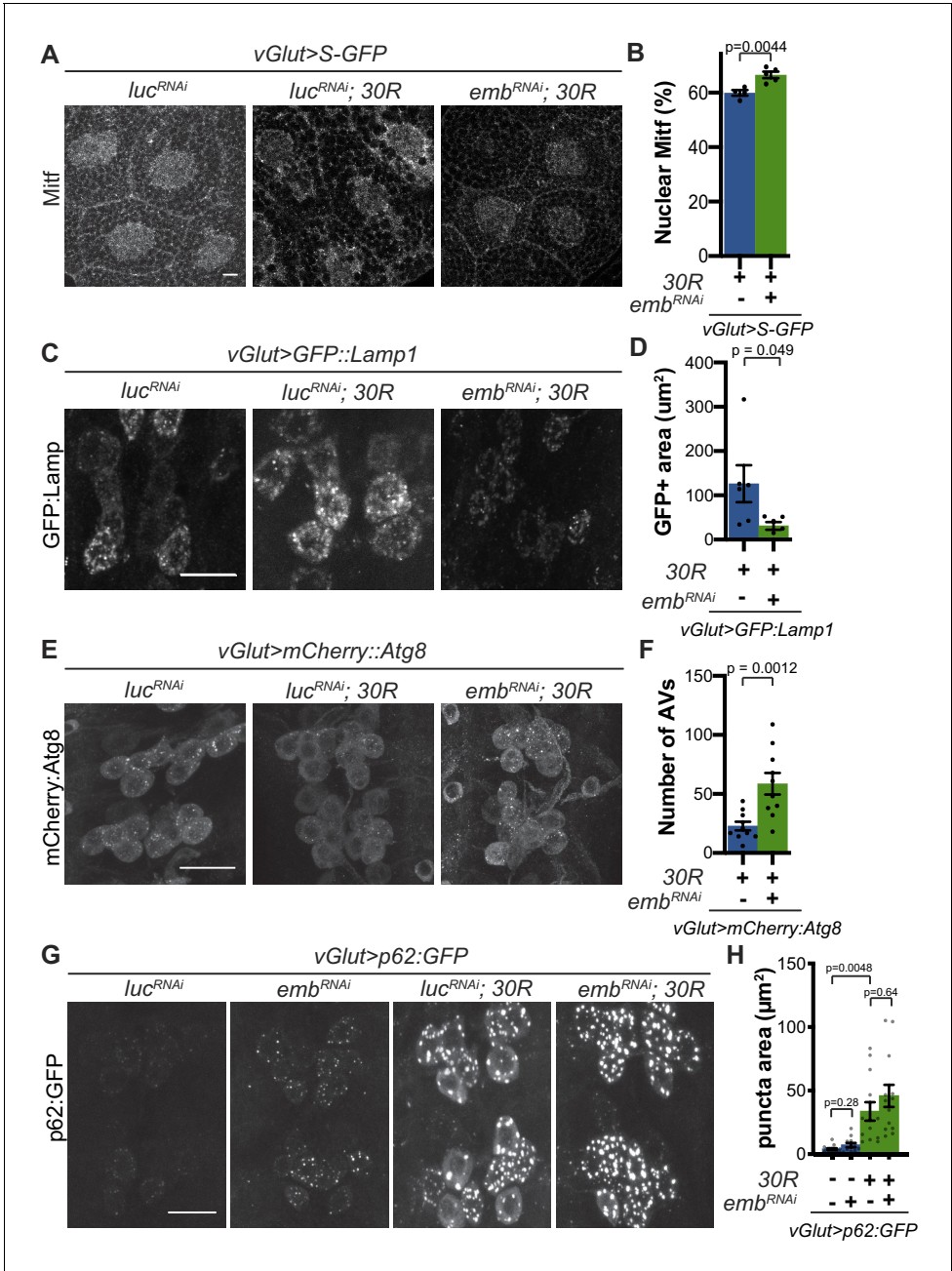

**Figure 5.** Modulation of nucleocytoplasmic transport rescues autophagolysosome dysfunction. (**A**) *Drosophila* larval salivary glands stained with anti-Mitf and DAPI expressing +/- *UAS-30R, UAS-shuttle-GFP* (S-GFP, not shown), and either control RNAi (*UAS-luc^{RNAi}*) or exportin RNAi (*UAS-emb^{RNAi}*) under the control of *vGlut-Gal4.* Scale bar = 10 μm (**B**) Quantification of percent (%) nuclear Mitf in A. Data are reported as mean ± SEM. Student's t-test, n = 4 and 5 larvae, respectively. (**C**) *Drosophila* motor neurons expressing *UAS-GFP:Lamp1* (N-terminal, luminal GFP) -/+ *UAS-30R* and *UAS-luc^{RNAi}* or exportin RNAi (*UAS-emb^{RNAi}*). Scale bar = 10 μm. (**D**) Quantification of C. Student's t-test, n = 6 larvae. (**E**) *Drosophila* motor neurons expressing *UAS-mCherry:Atg8* +/- *UAS-30R* and either control RNAi (*UAS-luc^{RNAi}*) or exportin RNAi (*UAS-emb^{RNAi}*). Scale bar = 10 μm. (**F**) Quantification of E. Data are reported as mean ± SEM. Mann-Whitney test, n = 10 larvae. (**G**) *Drosophila* motor neurons expressing *UAS-p62:GFP* -/+ *UAS-30R* and either control RNAi (*luc^{RNAi}*) or exportin RNAi (*emb^{RNAi}*) under the control of *vGlut-Gal4.* Scale bar = 10 μm. (**H**) Quantification of G. Data are reported as mean ± SEM. Brown-Forsythe and Welch ANOVA test, p<0.0001, followed by Dunnett's T3 multiple comparisons, n = 12–14 larvae per genotype.

The online version of this article includes the following figure supplement(s) for figure 5:

**Figure supplement 1.** Nucleocytoplasmic transport disruption is upstream of autophagic defects.

overexpressing the G4C2 repeats (*Figure 5—figure supplement 1C*), suggesting that NCT disruption is sufficient to disrupt autophagic flux in *Drosophila* motor neurons.

## Mitf is mislocalized and inactivated in *Drosophila* models of C9-ALS/FTD

Because we observed a reduction in autophagosomes and expansion of lysosome-related organelles, we hypothesized that transcription factors regulating autophagolysosomal function may be mislocalized to the cytoplasm due to disrupted nuclear import. The MiT/TFE family of transcription factors (TFEB, TFE3, MITF, and TFEC) regulates multiple steps of autophagy from autophagosome biogenesis through lysosome acidification via a network of genes called the Coordinated Lysosome Expression And Regulation (CLEAR) network (*Settembre et al., 2011*). These transcription factors are regulated by localization between the cytoplasm and nucleus (*Li et al., 2018*). In *Drosophila*, this conserved transcription factor family is represented by a single homolog called *Mitf* (*Bouché et al., 2016*; *Zhang et al., 2015b*). *Mitf* knockdown in the nervous system causes lysosomal defects similar to those observed in G4C2-expressing flies (*Bouché et al., 2016*; *Hallsson et al., 2004*; *Sardiello et al., 2009*; *Song et al., 2013*). Additionally, TFEB levels are reduced in superoxide dismutase 1 (*SOD1*) mutant cell culture and mouse ALS models (*Chen et al., 2015*) as well as in ALS and Alzheimer's patient brain tissue (*Wang et al., 2016*). Therefore, we hypothesized that impaired Mitf nucleocytoplasmic transport might underlie the autophagolysosomal phenotypes in fly models of C9-ALS. Indeed, both salivary gland cells and motor neurons expressing *30R* show a reduction in percent nuclear Mitf (*Figure 4A–D*). To assess whether disrupted Mitf NCT alters CLEAR gene expression in adult heads, we expressed 30R using a ubiquitous inducible driver, *daughterless-GeneSwitch (daGS)*. In control flies, a mild (~1.75 fold) overexpression of *Mitf* mRNA resulted in a significant upregulation of 3 of the 7 Mitf targets tested (the vesicular ATPase (v-ATPase) subunits *Vha16-1, Vha68-2, and Vha44*) and a trend towards upregulation of 4 others (*Figure 4E*). Importantly, co-expression of *30R* with *daGS >Mitf* led to a similar ~2 fold increase in *Mitf* transcripts but did not induce Mitf target genes (*Figure 4E*). This lack of Mitf target induction in *30R* flies suggests that decreased nuclear import of Mitf suppresses the ability of *30R*-expressing flies to upregulate CLEAR genes in order to maintain or induce autophagic flux.

We next examined whether rescue of nucleocytoplasmic transport defects in *30R*-expressing animals can rescue Mitf nuclear import and autophagolysosomal defects. Exportin-1 has recently been demonstrated to regulate Mitf/TFEB nuclear export (*Li et al., 2018*; *Silvestrini et al., 2018*). Knockdown of exportin-1 (*Drosophila emb*) rescues G4C2-mediated cytoplasmic Mitf mislocalization in the salivary gland (*Figure 5A–B*) and GFP:Lamp accumulation in motor neurons (*Figure 5C–D*). Importantly, *emb* knockdown increases the total number of autophagosomes in G4C2-expressing motor neuron cell bodies by ~3 fold (*Figure 5E–F*), suggesting that nuclear retention of Mitf rescues autophagolysosomal defects. However, *emb* knockdown caused a slight elevation of p62:GFP puncta intensity in controls and did not rescue the accumulations of p62:GFP in 30R-expressing motor neurons (*Figure 5G–H*). Together, these data indicate that autophagolysosomal dysfunction in *30R*-expressing animals occurs downstream of nucleocytoplasmic transport disruption, whereas inhibition of nuclear export is not sufficient to rescue p62 accumulation.

## Mitf rescues G4C2 repeat-mediated degeneration

Since Mitf mislocalization contributes to autophagolysosome defects in a fly C9-ALS model, we hypothesized that increasing total levels of Mitf might compensate for impaired nuclear import. While high level *Mitf* overexpression is toxic in *Drosophila* (*Hallsson et al., 2004*), a genomic duplication construct containing the *Mitf* gene lacking the DNA repetitive intron 1 (*Mitf Dp*) (*Zhang et al., 2015b*), is sufficient to partially rescue *30R*-mediated eye degeneration, while *Mitf* knockdown enhances eye degeneration (*Figure 6A–B*). Furthermore, pupal lethality caused by *30R* expression in motor neurons and climbing impairment in *elavGS >30R* flies are also partially rescued by *Mitf Dp* (*Figure 6C–D*). In contrast, *Mitf Dp* did not rescue the severe rough eye phenotype observed with *GMR-Gal4* overexpression of poly(GR)$_{36}$ (*Figure 6—figure supplement 1A–B*), suggesting that *Mitf Dp* rescues toxicity caused by the G4C2 repeat RNA rather than the DPRs alone. To determine whether increased levels of Mitf rescue G4C2-mediated neurodegeneration through effects on the autophagolysosomal pathway, we examined GFP:Lamp and p62:GFP expression in

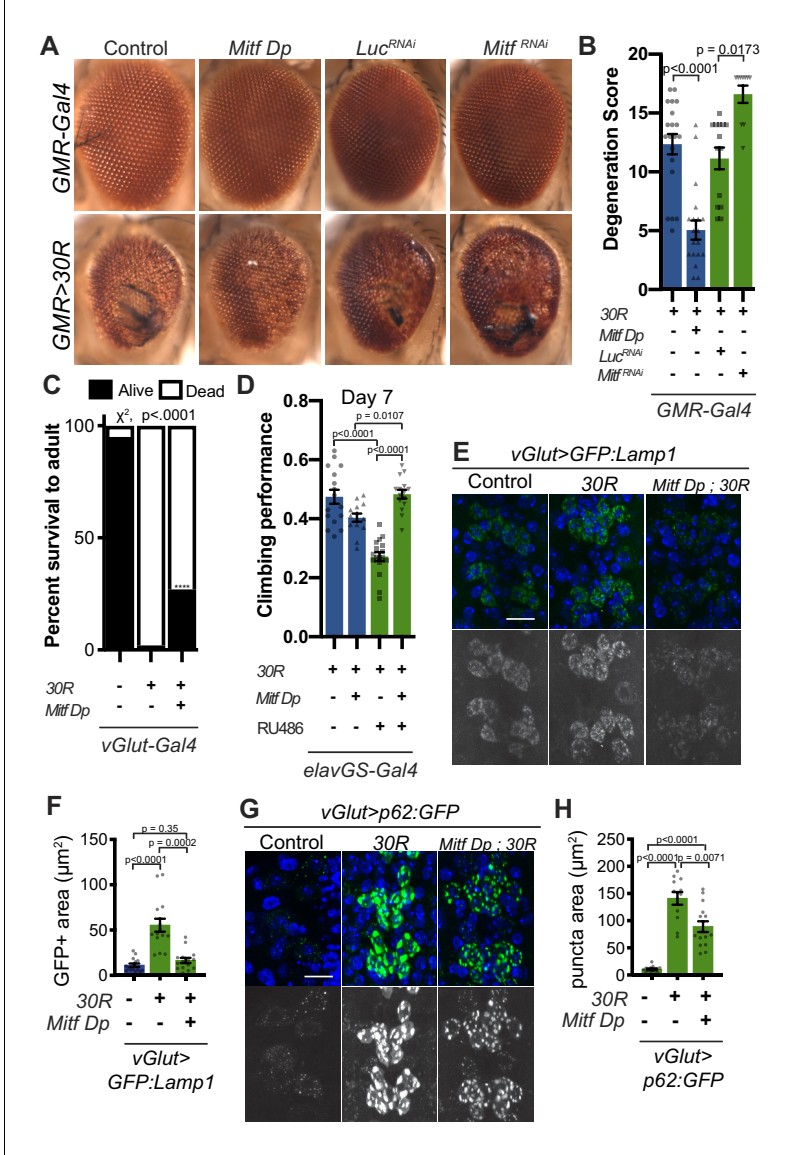

**Figure 6.** Transcription factor Mitf/TFEB suppresses neurodegeneration caused by G4C2 expansion via lysosome activity. (A) 15-day-old *Drosophila* eyes expressing *UAS-30R* under the control of *GMR-Gal4*, crossed to controls (*w1118* or *UAS-luciferase RNAi*), genomic *Mitf Duplication* (*Mitf Dp*), or *UAS-Mitf RNAi*. (B) Quantification of external eye degeneration shown in A. Data are reported as mean ± SEM. Kruskal-Wallis test, p<0.0001, followed by Dunn's multiple comparisons, n = 10–20 adults per genotype. (C) Percent of pupal eclosion in *Drosophila* expressing *UAS-30R* under the control of *vGlut-Gal4* -/+ *Mitf Dp* compared to *vGlut-Gal4/w1118* control. Fisher's exact test, n = 133, 139, and 84 pupae, respectively. (D) Adult *Drosophila* expressing *UAS-30R* under the control of the inducible, pan-neuronal *elavGS* driver induced with 200 μM RU486 have decreased climbing ability at 7 days of age. Co-expressing *Mitf Dp* with *UAS-30R* rescues climbing ability. One-way ANOVA, p<0.0001, followed by Sidak's multiple comparisons, n = 14–17 groups of 10 flies per genotype. (E) Representative images of motor neurons expressing *UAS-GFP:Lamp1* for control (*w1118*), *UAS-30R*, or coexpressing *Mitf Dp* and *UAS-30R*. Scale bar = 10 μm (F) Quantification of the GFP positive (GFP+) area of GFP:Lamp1 in E. Data are reported as mean ± SEM. Brown-Forsythe and Welch ANOVA, p<0.0001, test followed by Dunnett's T3 multiple comparisons, n = 15 per genotype. (G) Representative images of motor neurons coexpressing *UAS-p62:GFP* with no repeats (control, *w1118*), *UAS-30R*, and *Mitf Dp* with *UAS-30R*. Scale bar = 10 μm. (H) Quantification of p62:GFP GFP+ puncta area in F. Data are reported as mean ± SEM. Brown-Forsythe and Welch ANOVA test, p<0.0001, followed by Dunnett's T3 multiple comparisons, n = 12–14 larvae per genotype.

The online version of this article includes the following figure supplement(s) for figure 6:

**Figure supplement 1.** Genetic increase of lysosome function rescues degeneration caused by G4C2 expression but not by poly-GR.

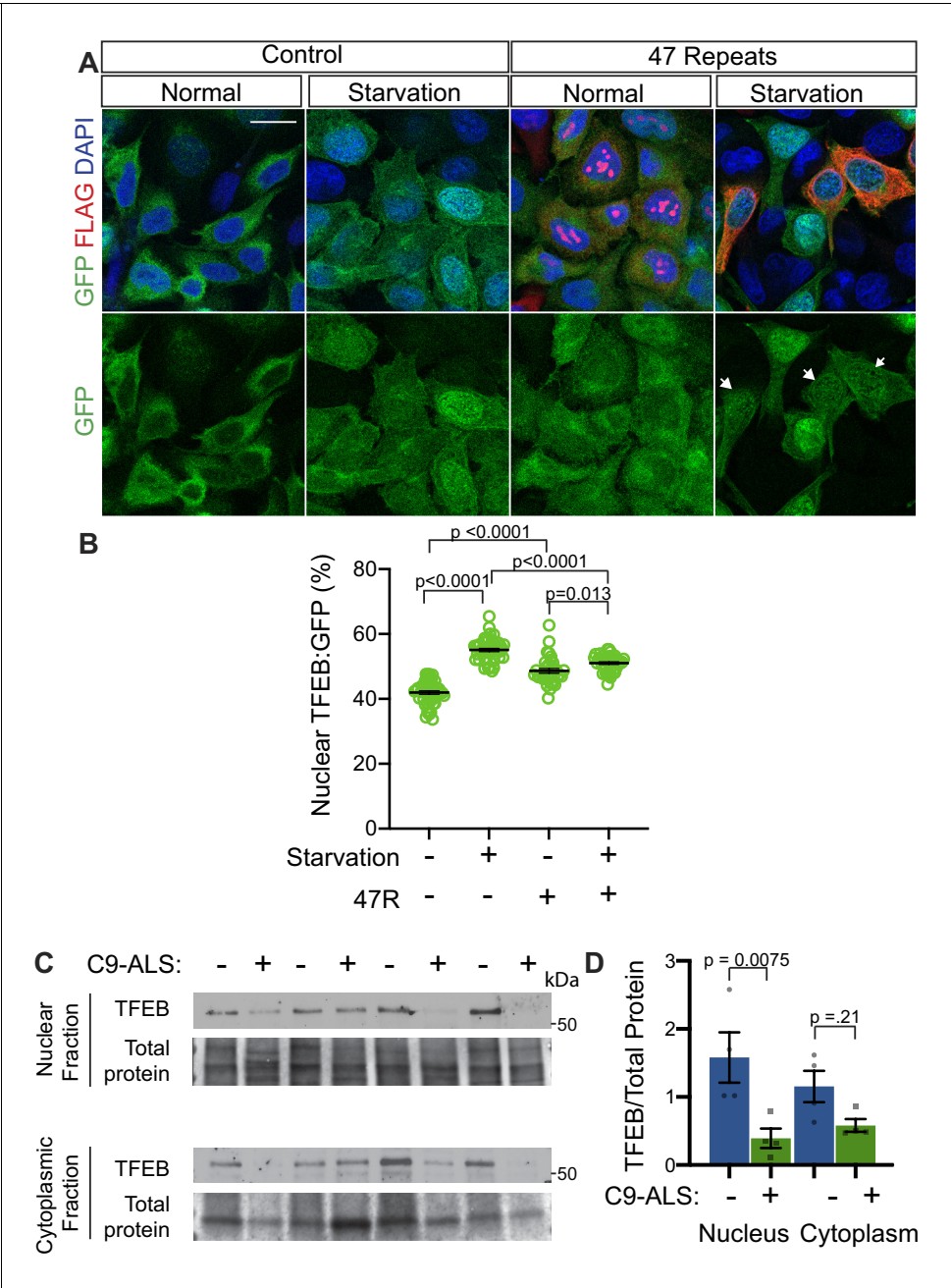

**Figure 7.** Nuclear TFEB is reduced in human cells expressing GGGGCC repeats and in C9-ALS human motor cortex. (**A**) HeLa cells stably expressing TFEB:GFP transfected with 0R (Control) or a 47R construct (Flag tag in frame with poly-GR) in normal media (DMEM) or starved (3 hr in EBSS) conditions. White arrowheads indicate transfected cells in the 47R starved group. (**B**) Quantification of cells from A showing the percent (%) nuclear TFEB:GFP (nuclear/total) for each group. Data are presented as mean ± SEM. One-way ANOVA, p<0.0001, with Sidak's multiple comparisons, n = 47, 47, 35, and 38 cells. (**C**) Western blot for TFEB of human motor cortex samples fractionated into cytoplasmic and nuclear samples from postmortem control and C9-ALS patient brains. (**D**) Quantification of TFEB levels against total protein loading (Faststain) in control and C9-ALS patients. Data reported are mean ± SEM. One-way ANOVA, p=0.0142, with Sidak's multiple comparisons, n = 4.

The online version of this article includes the following figure supplement(s) for figure 7:

**Figure supplement 1.** DPRs affect TFEB import in HeLa Cells.

*30R*-expressing motor neurons. Indeed, *Mitf Dp* rescues increased GFP:Lamp1 expression (*Figure 6E–F*) and reduces p62:GFP accumulation in motor neurons of *vGlut >30R* larvae (*Figure 6G–H*). Thus, increasing Mitf levels in multiple neuronal subtypes in *Drosophila* suppresses G4C2-mediated neurotoxicity, consistent with our hypothesis that loss of nuclear *Mitf* is a key contributor to G4C2-mediated neurodegeneration.

If the impaired lysosomal function we observe in our *Drosophila* model is contributing to neurodegeneration downstream of NCT defects, we would predict that genetic upregulation of key regulators of lysosome function may suppress degenerative phenotypes. Indeed, overexpression of *Rab7*, the small GTPase required for fusion of autophagosomes with lysosomes, or *Trpml*, a lysosomal calcium channel, suppress eye degeneration (*Figure 6—figure supplement 1C–D*). Furthermore, overexpression of key lysosomal v-ATPase subunits whose expression is regulated by *Mitf* also suppresses neurodegeneration in the *Drosophila* eye, while RNAi-mediated knockdown enhances degeneration (*Figure 6—figure supplement 1C–D*). Interestingly, loss of the ALS-associated gene *ubqn* in *Drosophila* was also rescued by increase in key lysosomal v-ATPase subunits or by nanoparticle mediated lysosome acidification (*Şentürk et al., 2019*). Overexpression of these Mitf-regulated genes also showed partial rescue of pupal lethality in animals expressing *30R* in motor neurons

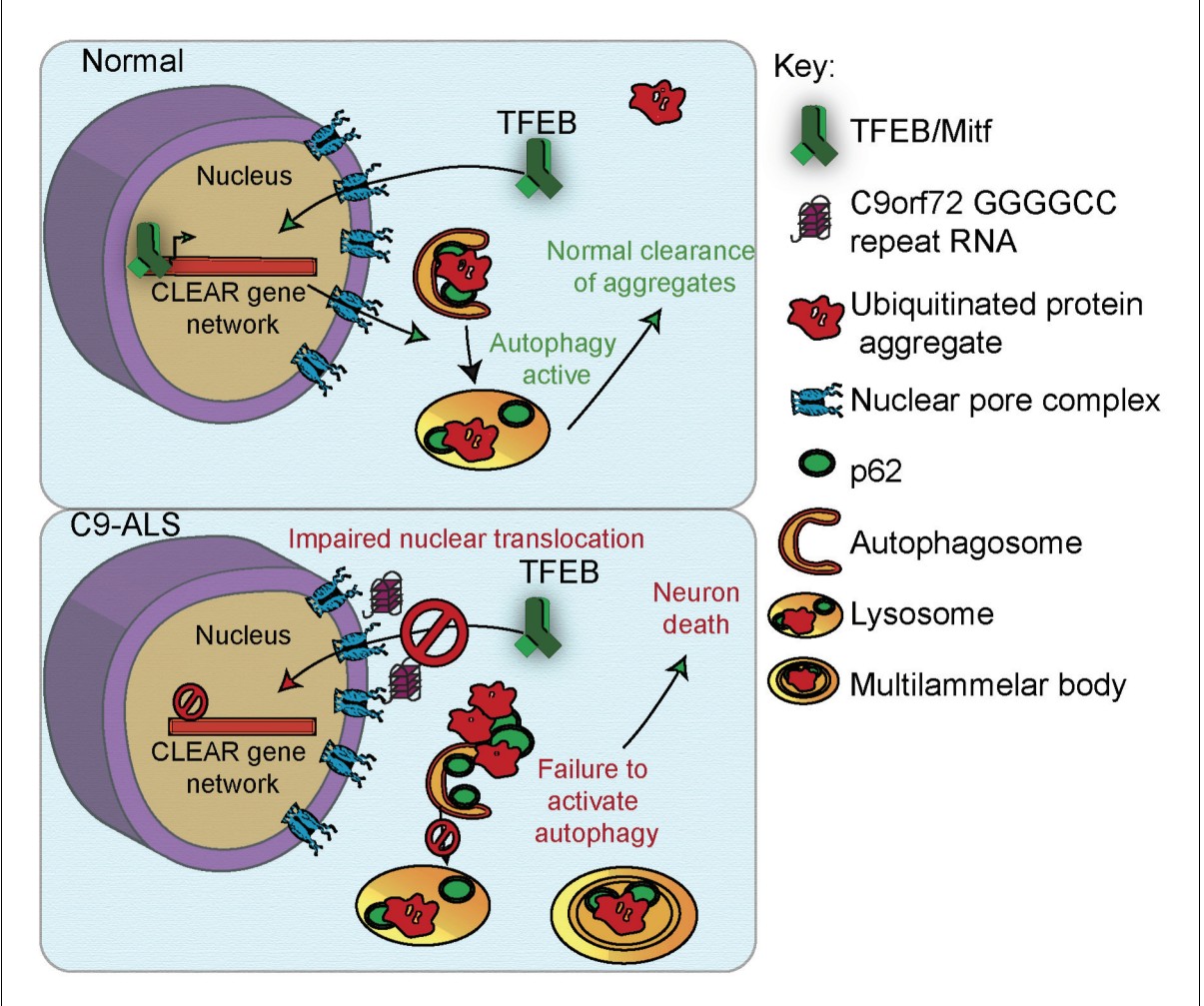

**Figure 8.** A proposed model of GGGGCC repeat expansion pathogenesis. G4C2 repeat expansion causes nucleocytoplasmic transport disruption through multiple proposed mechanisms including G4C2 RNA binding of RanGAP and stress granule recruitment of nucleocytoplasmic transport machinery. Transport disruption leads to a blockage in the translocation of autophagy-mediating transcription factors such as Mitf/TFEB to the nucleus in response to proteotoxic stress. Failure to induce autophagic flux leads to autophagy pathway disruption such as the accumulation of large, non-degradative lysosomes and MLBs. Loss of autophagic flux leads to accumulation of Ref(2)P/ p62 and ubiquitinated protein aggregates, leading to chronic protein stress signaling and eventually neuronal cell death.

(*Figure 6—figure supplement 1E*). These findings suggest a model whereby downregulation or cytoplasmic retention of Mitf targets leads to lysosomal disruption in G4C2-repeat-expressing flies.

## Nuclear TFEB is reduced in human cells and motor cortex with GGGGCC repeat expansions

In humans, TFEB is the homolog of *Drosophila Mitf* that is best characterized for its role in autophagy and has been implicated in neurodegenerative disease (*Cortes and La Spada, 2019*; *Martini-Stoica et al., 2016*). Interestingly, a previous study showed nuclear TFEB was selectively depleted in the motor cortex of a sample of five ALS patients compared to five controls (*Wang et al., 2016*). To test the relevance of our findings in *Drosophila* models to human disease, we next examined whether G4C2 repeat expression impairs nuclear import of TFEB in HeLa cells stably expressing TFEB:GFP (*Roczniak-Ferguson et al., 2012*) using a 47-repeat (47R) G4C2 construct that expresses tagged DPRs (see Materials and methods). In control cells, TFEB:GFP is predominantly localized to the cytoplasm, whereas induction of autophagy by 3 hr starvation leads to robust nuclear translocation of TFEB (*Figure 7A–B*). In contrast, while 47R-expressing cells have a mild basal elevation of nuclear TFEB, the nuclear translocation of TFEB in response to starvation is significantly impaired relative to control cells (*Figure 7A–B*). We then tested the effect of expression of DPRs produced by alternate codons (i.e. in the absence of G4C2 repeats): poly-glycine-alanine (poly-GA$_{50}$), poly-glycine-arginine (poly-GR$_{50}$), and poly-proline-arginine (poly-PR$_{50}$) (*Figure 7—figure supplement 1A–B*). While poly-GA$_{50}$ causes a mild decrease in TFEB nuclear translocation, expression of poly-GR$_{50}$ or poly-PR$_{50}$ does not disrupt TFEB:GFP nuclear translocation. These data suggest that human cells expressing an expanded G4C2 repeat, but not DPRs, are unable to efficiently import TFEB into the nucleus in response to stimuli.

To further investigate the relevance of loss of TFEB nuclear import to C9-ALS patients, we obtained human motor cortex samples from four non-neurological controls and four C9-ALS patients (*Supplementary file 2*). These samples were fractionated into cytoplasmic and nuclear-enriched fractions and assayed for TFEB using Western analysis. TFEB is reduced by an average of 76% in the nuclear fraction and by about 50% in the cytoplasm in C9-ALS compared to controls (*Figure 7C–D*, *Figure 7—figure supplement 1C*). These data suggest that TFEB protein is downregulated in C9-ALS/FTD motor cortex, but the greatest depletion occurs in the nucleus. Therefore, we propose a model whereby disruption of protein nuclear import by the *C9orf72*-HRE results in a failure of Mitf/TFEB to translocate to the nucleus to regulate the autophagic response to protein stress (*Figure 8*).

## Discussion

Our work has revealed that the ALS-associated G4C2 hexanucleotide repeat is sufficient to disrupt multiple aspects of autophagy. In *Drosophila*, G4C2 repeats cause loss of autophagosomes and disrupt lysosomal structure and function. This accumulation of autolysosomes and lysosome-related organelles (MLBs) has been observed in lysosomal storage disorders and has been reported in spinal cord tissue from sporadic ALS patients (*Bharadwaj et al., 2016*; *Parkinson-Lawrence et al., 2010*; *Sasaki, 2011*). Regulation of protein and lipid homeostasis by the lysosome may be particularly important in neurons since they are post-mitotic and have high energy demands (*Fraldi et al., 2016*). Loss of function of *C9orf72* also disrupts autophagy and lysosomal function in multiple cell types (*Farg et al., 2014*; *Ji et al., 2017*; *O'Rourke et al., 2015*; *Sellier et al., 2016*; *Shi et al., 2018*; *Sullivan et al., 2016*; *Ugolino et al., 2016*; *Webster et al., 2016*; *Yang et al., 2016*; *Zhu et al., 2020*), suggesting a mechanism whereby G4C2 repeats may have synergistically detrimental effects with haploinsufficient *C9orf72* in C9-ALS/FTD patients. Additionally, multiple forms of familial ALS are caused by mutations in genes in autophagy and lysosome function (*Evans and Holzbaur, 2019*; *Lin et al., 2017*; *Ramesh and Pandey, 2017*), and upregulation of lysosome function has been proposed to be beneficial in multiple preclinical models of ALS (*Donde et al., 2020*; *Mao et al., 2019*; *Şentürk et al., 2019*; *Shi et al., 2018*). Thus, our findings suggest that, as has been shown in other forms of ALS, neurotoxicity of G4C2 repeats in C9 ALS-FTD is at least partially caused by disrupted autophagolysosomal function.

The finding that *ref(2)P* knockdown prevents or delays G4C2-mediated neurodegeneration is surprising, as p62/SQSTM1 is thought to link toxic ubiquitinated aggregates to LC3 to remove aggregates via selective autophagy (*Cipolat Mis et al., 2016*; *Levine and Kroemer, 2008*; *Saitoh et al.,*

*2015*). However, other studies have also suggested that p62 may contribute to (rather than ameliorate) toxicity of ubiquitinated proteins. For example, *Atg7*$^{-/-}$ mice display severe defects in autophagy and accumulation of p62-positive protein aggregates in the liver and brain, and knockout of p62 in these mice prevents the formation of ubiquitinated aggregates and rescues liver dysfunction via suppression of chronic oxidative stress signaling (*Komatsu et al., 2007*). Additionally, Ataxia Telangiectasia Mutated-mediated DNA double stranded break repair is impaired in cultured neurons expressing the *C9orf72*-HRE, and this phenotype is rescued by p62 knockdown (*Walker et al., 2017*). These findings suggest that increases in p62 may contribute to DNA damage previously described in C9-ALS. Further, p62 is found to co-localize with DPRs in C9-ALS patients (*Al-Sarraj et al., 2011*; *Mackenzie et al., 2014*; *Mori et al., 2013*) and may promote protein aggregation. We hypothesize that p62-positive aggregate or oligomer formation in C9-patients contributes to neurotoxicity by activating downstream signaling pathways that are alleviated by autophagy-mediated clearance.

While many groups have reported nucleocytoplasmic transport dysfunction in ALS, it has remained unclear how NCT disruption causes ALS. Stress granules can recruit nuclear pore proteins to the cytoplasm and cause nucleocytoplasmic transport defects, suggesting that the disruptions in phase separation of RNA-binding proteins may lie upstream of nucleocytoplasmic transport defects (*Zhang et al., 2018*). Recently, Ortega et al. discovered that hyperactivity of nonsense-mediated decay may lie downstream of nucleocytoplasmic transport, indicating that multiple proteostasis pathways may be disrupted (*Ortega et al., 2020*). Additionally, selective autophagy is required for nuclear pore turnover (*Lee et al., 2020*), implying that autophagy defects may contribute to the cytoplasmic nuclear pore pathology found in C9-ALS patients and animal models. Our data show that in *Drosophila*, HeLa cells, and human tissue, nucleocytoplasmic transport defects lead to an inability to activate TFEB translocation to the nucleus, causing widespread autophagy defects and accumulation of protein aggregates (*Figure 8*). Interestingly, genetic inhibition of nuclear export or increase in Mitf expression are able to strongly rescue autophagosome and lysosome phenotypes and neurodegeneration, but do not result in complete clearance of p62 accumulations (*Figures 5–6*). Additional studies will be needed to better understand the relationship between p62 accumulation, autophagy, nucleocytoplasmic transport, and neurodegeneration. Overall, these findings place nucleocytoplasmic transport defects in ALS upstream of proteostasis defects.

Importantly, TFEB has been previously proposed as a therapeutic target in ALS and other neurodegenerative disease (*Cortes and La Spada, 2019*). Upregulation of TFEB signaling helps clear multiple types of proteotoxic aggregates found in Alzheimer's disease, Parkinson's disease, Huntington's disease, ALS and FTD (*Decressac et al., 2013*; *Parr et al., 2012*; *Polito et al., 2014*; *Torra et al., 2018*; *Vodicka et al., 2016*). Our study suggests that modulation of TFEB nucleocytoplasmic transport may be an additional therapeutic target, and that targeting both nucleocytoplasmic transport and autophagy may act synergistically in ALS and FTD.

## Materials and methods

### *Drosophila* genetics

*Drosophila* were raised on standard cornmeal-molasses food at 25°C. For eye degeneration, *GMR-GAL4, UAS-30R*/CyO, *twi-GAL4, UAS-GFP* were crossed to *UAS-modifier* lines or background controls and *GMR-GAL4, UAS-30R/UAS-modifier* or *GMR-GAL4, UAS-30R/+* were selected (where *UAS-modifier* can be on any chromosome) from the offspring and aged at 25°C for 15 days. Eye degeneration is quantified using a previously described method (*Ritson et al., 2010*). Briefly, points were added if there was complete loss of interommatidial bristles, necrotic patches, retinal collapse, loss of ommatidial structure, and/or depigmentation of the eye. Eye images were obtained using a Nikon SMZ 1500 Microscope and Infinity 3 Luminera Camera with Image Pro Insight 9.1 software.

For pupal survival assay, either three males from *vGlut-Gal4* or *vGlut-Gal4; UAS-30R/TM6G80(Tb)* were crossed to 5–6 female flies containing UAS-modifier lines or background controls. Parental adult crosses were transferred to fresh vials every 2–3 days. After 15 days, non-tubby pupated flies that were (either *vGlut-Gal4/UAS-modifier*, *vGlut/+*; *UAS-30R*, or *vGlut-Gal4/UAS-modifier*; *UAS-30R*) were scored as either eclosed (empty pupal case) or non-eclosed (typically a fully developed pharate adult fly unable to eclose from pupal case due to paralysis).

For the climbing assay, *UAS-30R; elavGS* were crossed to experimental or genetic background controls. Adults were transferred 3–5 days after eclosion to vials containing 200 µM RU486 food or ethanol vehicle alone and transferred to new vials every 2–3 days. After aging 7–10 days, groups of 10 flies were placed into empty food vials and were tapped to the bottom and then locomotor function assessed by their negative geotaxis (flies reflexively crawl against gravity) response as measured by ability to climb 8 cm in 10 seconds. Each cohort of 10 flies was tested 10 times to obtain an average. N represents individual cohorts of 10 flies.

## *Drosophila* drug feeding

Cornmeal-molasses-yeast fly food was melted and then cooled for 5 min before being mixed with concentrations of mifepristone (RU486), rapamycin, or trehalose and cooled to room temperature. Ethanol or DMSO was used as a vehicle control. Parent flies were crossed on normal food, and then they were transferred to food containing drug every 2–3 days such that their offspring would develop in food containing drug or adult offspring were transferred to drug food once eclosed as noted. Wandering third-instar larvae were selected for immunostaining or western blot analysis. Adult flies were aged on the drug-containing food for 15 days before analyzing their eye morphology or assessed for climbing ability on the day noted.

## Quantitative RT–PCR

For each genotype, mRNA was collected from 5 flies or 30 heads using the TRIzol reagent following the manufacturer's protocol. Reverse transcription was performed using SuperScript III First-Strand synthesis kit following the manufacturer's protocol. Quantitative PCR was performed using SYBR Green PCR system on a 7900HT fast Real-Time PCR system (Applied Biosystem). The primers for G4C2 repeats were designed to amplify a 3' region immediately after the repeats in the UAS construct.

## Immunofluorescence staining and imaging

For *Drosophila* ventral nerve cords, wandering third-instar larvae were dissected in HL3 (*Stewart et al., 1994*) using a standard larval fillet dissection then fixed in 4% paraformaldehyde (or Bouin's fixative for *UAS-mCherry:Atg8* experiments) (Sigma) for 20 min, followed by wash and penetration with PBS 0.1% Triton X-100 (PBX) for 3 × 20 min washes. The tissues were blocked for 1 hr at room temperature in PBS with 5% normal goat serum (NGS) and 0.1% PBX, then stained with primary antibodies at 4C overnight (16 hr). Tissues were washed three times for 20 min each with 0.1% PBX. Secondary antibodies (Goat antibodies conjugated to Alexa Fluor 568, 488, 633) diluted in 0.1% PBX 5% NGS and incubated for 2 hr and then washed three times for 20 min each with 0.1% PBX. During one wash, DAPI was added to the prep at a final concentration of 1 µg/mL. Larvae were mounted in Fluoromount-G (Invitrogen).

*Drosophila* salivary glands were dissected using a standard protocol and stained as above excepting for stronger solubilization with 0.3% PBX. Fixed cells or tissues were analyzed under an LSM780 or LSM800 confocal microscope (Carl Zeiss) with their accompanying software using Plan Apochromat 63 ×, NA 1.4 DIC or Plan Apochromat 40×, 1.3 Oil DIC objectives (Carl Zeiss) at room temperature. Images were captured by an AxioCam HRc camera (Carl Zeiss) and were processed using ImageJ/Fiji. To quantify fluorescent intensities, after opening the images in ImageJ/Fiji, certain areas/bands were circled and the intensities were measured. Puncta were counted using the Analyze Particles function in Image J using the same thresholding across experiments. Images are representative and experiments were repeated two to five times.

## Western blotting

Tissues or cells were homogenized and/or lysed in RIPA buffer (50 mM Tris-HCl pH 7.4, 150 mM NaCl, 0.1% SDS, 0.5% sodium deoxycholate, and 1% Triton X-100) supplemented with protease inhibitor cocktail (Complete, Roche) using microcentrifuge pestles, and then were incubated in RIPA buffer on ice for 20 min. Samples were spun down at 100 g for 5 min to remove carcass and unbroken cells. For protein quantification, solution was diluted and measured by BCA assay (Thermo Fischer Scientific).

For nucleocytoplasmic fractionation of autopsy tissue, fractionation was performed with the NE-PER Nuclear and Cytoplasmic Extraction Kit according to the manufacturer's protocol. For detection of proteins in the whole fraction, *Drosophila* larvae were solubilized in 8M urea. For the soluble and pelleted fraction, larvae were first solubilized in RIPA buffer as described above. The samples were spun down at 15000 rpm for 20 min and the soluble supernatant was set aside. Freshly prepared 8M urea buffer (Sigma) was added to the pellet and dissolved through vortexing. Samples were spun again at 15000 rpm for 20 min and urea-soluble pellet fraction was collected. A small amount of sample buffer dye was added and urea-buffered protein samples were run immediately on SDS-PAGE without heating. For immunoblot, 10–50 μg of total protein sample was mixed with 4x Laemmli buffer (Bio-Rad) and heated at 98°C for 10 min. The protein samples were run on 4–15% SDS Mini-PROTEAN TGX Precast Gels (Bio-Rad) and transferred to nitrocellulose membrane. TBST (50 mM Tris-HCl pH 7.4, 1% Triton X-100) with 5% non-fat milk (Bio-Rad) was used for blocking.

## Electroretinogram (ERG) Assay

For ERG recordings, *Rh1-GAL4/UAS-LacZ* and *Rh1-GAL4/UAS-30*R flies were aged at 25°C in 12 hr light/12 hr dark cycle. ERG recordings were performed as described (*Şentürk et al., 2019*). In brief, adult flies were immobilized on a glass slide by glue. A reference electrode was inserted in the thorax and a recording electrode was placed on the eye surface. Flies were maintained in the darkness for at least 2 min prior to 1 s flashes of white light pulses (LED source with daylight filter), during which retinal responses were recorded and analyzed using WinWCP (University of Strathclyde, Glasgow, Scotland) software. At least five flies were examined for each genotype and timepoint.

## Transmission Electron Microscopy (TEM)

*Rh1-GAL4/UAS-lacZ* and *Rh1-GAL4/UAS-30*R flies were aged at 25°C in 12 hr light/12 hr dark cycle. Retinae of adult flies were processed for TEM imaging as previously described (*Chouhan et al., 2016*). Three flies were examined for each genotype and timepoint.

## Plasmids Source and Construction

pSF-CAG-Amp (0G504) was purchased from Oxford Genetics. We generated a mammalian expression plasmid pSF-(G4C2)47-VFH (V5-Flag-His), which can express 47 G4C2 repeats with three different tags to monitor expression of DPRs (polyGP-V5, polyGA-His, and polyGR-Flag). pEGFP-(GA,GR, or PR)50 was obtained from Davide Trotti (*Wen et al., 2014*), and the GFP cDNA sequence was replaced with mCherry by digesting with BamHI and XhoI.

## TFEB:GFP HeLa cell culture, transfection, and immunofluorescence analysis

HeLa cell line with stable expressing TFEB:GFP was a gift from Dr. Shawn Ferguson at Yale University. Hela cells were grown in DMEM media (Invitrogen) supplemented with 10% fetal bovine serum (Hyclone Laboratories Inc). The cell line was authenticated by observing nuclear translocation of TFEB:GFP in the presence of starvation (*Figure 7*). Absence of mycoplasma contamination was confirmed by staining with DAPI. Transfection was performed using Lipofectamine 2000 (Invitrogen) according to the manufacturer's instructions. Briefly, 1–2 μg of cDNA was diluted into 100 μl of Opti-MEM I Medium (Invitrogen) and mixed gently. Lipofectamine 2000 mixture was prepared by diluting 2–4 μl of Lipofectamine 2000 in 100 μl of Opti-MEM I Medium. The ratio of DNA to Lipofectamine 2000 used for transfection was 1:two as indicated in the manual. The DNA-Lipofectamine 2000 mixture was mixed gently and incubated for 20 min at room temperature. Cells were directly added to the 200 μl of DNA-Lipofectamine 2000 mixture. After 48 hr, transfected HeLa cells were treated with EBSS medium for 3 hr for starvation. HeLa cells were fixed with 4% PFA at room temperature for 15 min, washed three times with PBS, permeabilized for 10 min with 1% PBTX, washed another three times with PBS, and blocked for 1 hr at room temperature with 10% normal goat serum (Sigma) diluted in 0.1% PBTX. Cells were then incubated overnight at 4°C with primary antibody mouse anti-Flag antibody. After three washes in PBS (5 min each), cells were incubated for 1 hr at room temperature with secondary antibodies (goat anti-Alexa Fluor 568) diluted in the blocking solution. Cells were washed three times in PBS and mounted with Prolong Gold anti-fade reagent with DAPI (Cell Signaling).

## Collection of human autopsied tissue

Human autopsied tissue used for these data are described in detail in *Supplementary file 2*. The use of human tissue and associated decedents' demographic information was approved by the Johns Hopkins University Institutional Review Board and ethics committee (HIPAA Form five exemption, Application 11-02-10-01RD) and from the Ravitz Laboratory (UCSD) through the Target ALS Consortium.

## Statistics

All quantitative data were derived from independent experiments. Each n value representing biological replicates is indicated in the figure legends. Statistical tests were performed in Prism version 8.3.1 or Microsoft Excel 16.34 and were performed as marked in the figure legends. All statistical tests were two-sided. Results were deemed significant when the P value $\alpha$ = 0.05. No statistical methods were used to predetermine sample size. The investigators were not blinded during experiments.

## Acknowledgements

This work was supported by NINDS R01NS082563 (TEL), R01NS094239 (TEL and JDR), F31 NS100401 (KMC), ALSA (TEL, KZ and JDR), and Target ALS (TEL, KZ, JDR, and HJB). KMC is a recipient of the PEO Scholar Award. HJB is an Investigator of the Howard Hughes Medical Institute. We thank Francesca Pignoni, Udai Pandey, Peng Jin, Adrian Isaacs, Eric Baehrecke, Helmut Kramer, Francesca Pignoni, Gábor Juhász, Patrick Dolph, L Miguel Martins, the Bloomington *Drosophila* Stock Center (NIH P40ODO18537) and Vienna *Drosophila* Research Center for *Drosophila* lines and/or antibodies and Shawn Ferguson and Davide Trotti for cell lines and constructs. The Johns Hopkins NINDS Multiphoton Imaging Core (NS050274) provided imaging equipment and expertise.

## Additional information

### Competing interests

Hugo J Bellen: Reviewing editor, *eLife*. The other authors declare that no competing interests exist.

### Funding

| Funder | Grant reference number | Author |
| --- | --- | --- |
| National Institute of Neurological Disorders and Stroke | R01NS082563 | Thomas E Lloyd |
| Amyotrophic Lateral Sclerosis Association | 17-IIP-370 | Thomas E Lloyd |
| National Institutes of Health | P40OD018537 | Hugo J Bellen |
| Howard Hughes Medical Institute | | Hugo J Bellen |
| National Institute of Neurological Disorders and Stroke | R01NS094239 | Jeffrey D Rothstein Thomas E Lloyd |
| National Institute of Neurological Disorders and Stroke | P30NS050274 | Thomas E Lloyd |
| National Institute of Neurological Disorders and Stroke | F31NS100401 | Kathleen M Cunningham |
| ALSA | | Jeffrey D Rothstein Ke Zhang Thomas E Lloyd |

The funders had no role in study design, data collection and interpretation, or the decision to submit the work for publication.

## Author contributions
Kathleen M Cunningham, Conceptualization, Resources, Data curation, Formal analysis, Supervision, Funding acquisition, Validation, Investigation, Visualization, Methodology, Writing - original draft, Project administration, Writing - review and editing; Kirstin Maulding, Zhongyuan Zuo, Helen Song, Junli Gao, Sandeep Dubey, Investigation; Kai Ruan, Formal analysis, Investigation, Visualization, Writing - review and editing; Mumine Senturk, Conceptualization, Investigation, Visualization; Jonathan C Grima, Resources, Methodology; Hyun Sung, Investigation, Visualization; Jeffrey D Rothstein, Supervision; Ke Zhang, Conceptualization, Supervision, Funding acquisition, Investigation, Methodology, Writing - review and editing; Hugo J Bellen, Conceptualization, Resources, Supervision, Investigation, Methodology, Writing - review and editing; Thomas E Lloyd, Conceptualization, Supervision, Funding acquisition, Investigation, Visualization, Methodology, Writing - original draft, Project administration, Writing - review and editing

## Author ORCIDs
Kathleen M Cunningham (iD) https://orcid.org/0000-0002-1347-9087
Kirstin Maulding (iD) https://orcid.org/0000-0002-2012-9747
Ke Zhang (iD) http://orcid.org/0000-0002-4794-8355
Hugo J Bellen (iD) http://orcid.org/0000-0001-5992-5989
Thomas E Lloyd (iD) https://orcid.org/0000-0003-4756-3700

## Ethics
Human subjects: The use of human tissue and associated decedents' demographic information was approved by the Johns Hopkins University Institutional Review Board and ethics committee (HIPAA Form 5 exemption, Application 11-02-10-01RD) and from the Ravitz Laboratory (UCSD) through the Target ALS Consortium.

## Decision letter and Author response
Decision letter https://doi.org/10.7554/eLife.59419.sa1
Author response https://doi.org/10.7554/eLife.59419.sa2

# Additional files
## Supplementary files
• Source data 1. Source data for all figures.

• Supplementary file 1. Candidate Screen of autophagy-related genes. Flies expressing 30 G4C2 repeats in the eye under control of GMR-GAL4 were crossed to the indicated UAS line and scored for enhancement (<0) or suppression (>0) as described (*Zhang et al., 2015a*).

• Supplementary file 2. Demographics of human patients. Motor cortex from postmortem brain autopsies from four C9-ALS patients and four non-neurological disease controls were used is this study (*Figure 7*). Patient ID, cause of death (diagnosis), gender, age at death, post-mortem interval (PMI) in hours, and presence of C9orf72 expanded hexanucleotide repeat are indicated.

• Transparent reporting form

## Data availability
All data generated or analysed during this study are included in the manuscript and supporting files.

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

# Appendix 1

**Appendix 1—key resources table**

| Reagent type (species) or resource | Designation | Source or reference | Identifiers | Additional information |
|---|---|---|---|---|
| Genetic reagent (D. melanogaster) | GMR-Gal4 | Bloomington *Drosophila* Stock Center | BDSC:1104 | w*; P{GAL4-ninaE.GMR}12 |
| Genetic reagent (D. melanogaster) | 30R | Peng Jin (**Xu et al., 2013**) | FlyBase: FBal0294759 | w[1118];UAS-(G$_4$C$_2$)$_{30}$ |
| Genetic reagent (D. melanogaster) | TRiP background control | Bloomington *Drosophila* Stock Center | BDSC: 36303 | y[1] v[1]; P{y[+t7.7]=CaryP}attP2 |
| Genetic reagent (D. melanogaster) | UAS-ref(2)P$^{RNAi\#1}$ | Bloomington *Drosophila* Stock Center | BDSC: 36111 | y[1] sc[*] v[1] sev[21]; P{y[+t7.7] v[+t1.8]=TRiP. HMS00551}attP2 |
| Genetic reagent (D. melanogaster) | UAS-ref(2)P$^{RNAi\ \#2}$ | Bloomington *Drosophila* Stock Center | BDSC: 33978 | y[1] sc[*] v[1] sev[21]; P{y[+t7.7] v[+t1.8]=TRiP. HMS00938}attP2 |
| Genetic reagent (D. melanogaster) | UAS-ref(2)P-HA | L.M. Martins (**de Castro et al., 2013**) | Flybase: FBtp0089618 | |
| Genetic reagent (D. melanogaster) | vGlut-Gal4 | Bloomington *Drosophila* Stock Center | Flybase: FBal0194519 | w[1118]; P{w[+mW.hs] =GawB}VGlut[OK371] |
| Genetic reagent (D. melanogaster) | elavGS | Adrian Isaacs | Flybase: FBtp0015149 | w[*]; P{elav-Switch.O} GSG301 |
| Genetic reagent (D. melanogaster) | UAS-poly (GR)$_{36}$ | Adrian Isaacs (**Mizielinska et al., 2014**) | BDSC: 58692 | w[1118]; P{{y[+t7.7] w[+mC] =UAS poly-GR.PO-36}attP40 |
| Genetic reagent (D. melanogaster) | Act-Gal4 | Bloomington *Drosophila* Stock Center | Flybase: FBti0183703 | y[1] w[*]; P{Act5C-GAL4} 17bFO1/TM6B, Tb1 |
| Genetic reagent (D. melanogaster) | UAS-ref(2)P: GFP | Thomas Neufeld (**Chang and Neufeld, 2009**) | Flybase: FBtp0041098 | |
| Genetic reagent (D. melanogaster) | UAS-mCherry-Atg8 | Bloomington *Drosophila* Stock Center | BDSC: 37749 | y[1] w[1118]; P{w[+mC]=UASp-GFP-mCherry-Atg8a}2 |
| Genetic reagent (D. melanogaster) | UAS-GFP: Lamp1 | Helmut Kramer (**Pulipparacharuvil et al., 2005**) | Flybase: FBtp0041063 | w[*]; P{w[+mC]=UAS-GFP-LAMP}2 |

*Continued on next page*

*Appendix 1—key resources table continued*

| Reagent type (species) or resource | Designation | Source or reference | Identifiers | Additional information |
|---|---|---|---|---|
| Genetic reagent (*D. melanogaster*) | UAS-3R; UAS-(G₄C₂)₃ | Adrian Isaacs (**Mizielinska et al., 2014**) | BDSC: 58687 | *w[1118]; P{{y[+t7.7] w[+mC]=UAS GGGGCC.3}attP40* |
| Genetic reagent (*D. melanogaster*) | UAS-36R; UAS-(G₄C₂)₃₆ | Adrian Isaacs (**Mizielinska et al., 2014**) | BDSC: 58688 | *w[1118]; P{{y[+t7.7] w[+mC]=UAS GGGGCC.36}attP40* |
| Genetic reagent (*D. melanogaster*) | UAS-44R; UAS-LDS(G₄C₂)₄₄ | Nancy Bonini (**Goodman et al., 2019b**) | BDSC: 84723 | *w[1118]; P{w[+mC]=UAS-LDS-(G4C2)44.GR-GFP}9* |
| Genetic reagent (*D. melanogaster*) | UAS-LacZ | Bloomington *Drosophila* Stock Center | BDSC: 3956 | *w[1118]; P{w[+mC]=UAS-lacZ.NZ}J312* |
| Genetic reagent (*D. melanogaster*) | Rh1-Gal4 | Bloomington *Drosophila* Stock Center | BDSC: 8961 | *P{ry[+t7.2]=rh1 GAL4}3, ry[506]* |
| Genetic reagent (*D. melanogaster*) | gRab7-YFP | Bloomington *Drosophila* Stock Center | BDSC: 62545 | *w[1118]; TI{TI}Rab7[EYFP]* |
| Genetic reagent (*D. melanogaster*) | UAS-Rab7-GFP | Bloomington *Drosophila* Stock Center | BDSC: 42706 | |
| Genetic reagent (*D. melanogaster*) | UAS-luciferase^RNAi | Bloomington *Drosophila* Stock Center | BDSC: 31603 | *y[1] v[1]; P{y[+t7.7] v[+t1.8]=TRiP. JF01355}attP2* |
| Genetic reagent (*D. melanogaster*) | UAS-S-GFP; | Bloomington *Drosophila* Stock Center | BDSC: 7032 | *w[1118]; P{w[+mC]=UAS-NLS-NES [+]-GFP}5A* |
| Genetic reagent (*D. melanogaster*) | UAS-RanGAP | **Zhang et al., 2015a** | | |
| Genetic reagent (*D. melanogaster*) | UAS-RanGAP^RNAi; | Bloomington *Drosophila* Stock Center | BDSC: 29565 | *y[1] v[1]; P{y[+t7.7] v[+t1.8]=TRiP.JF03244}attP2 /TM3, Sb[1]* |
| Genetic reagent (*D. melanogaster*) | UAS-CD8:GFP | Bloomington *Drosophila* Stock Center | Flybase: FBti0012685 | *y[1] w[*]; P{w[+mC] =UAS-mCD8::GFP.L}LL5* |
| Genetic reagent (*D. melanogaster*) | UAS-Mitf-HA | Francesca Pignoni (**Zhang et al., 2015b**) | | |
| Genetic reagent (*D. melanogaster*) | daGS | Bloomington *Drosophila* Stock Center | Flybase: FBtp0057039 | *w[*]; P{w[+mC]=da-GSGAL4.T}* |

*Continued on next page*

*Appendix 1—key resources table continued*

| Reagent type (species) or resource | Designation | Source or reference | Identifiers | Additional information |
|---|---|---|---|---|
| Genetic reagent (*D. melanogaster*) | *UAS-embargoed*$^{RNAi}$ | Bloomington *Drosophila* Stock Center | BDSC: 31353 | *y[1] v[1]; P{y[+t7.7] v[+t1.8]=TRiP. JF01311}attP2* |
| Genetic reagent (*D. melanogaster*) | *Mitf duplication; Mitf*$^{SI-RES}$ | Francesca Pignoni (**Zhang et al., 2015b**) | Flybase: FBtp0115483 | |
| Genetic reagent (*D. melanogaster*) | *UAS-Mitf*$^{RNAi}$ | Bloomington *Drosophila* Stock Center | BDSC: 43998 | *y[1] sc[*] v[1] sev[21]; P{y[+t7.7] v[+t1.8]=TRiP. HMS02712}attP2* |
| Genetic reagent (*D. melanogaster*) | *UAS-Rab7*$^{WT}$ | Bloomington *Drosophila* Stock Center | BDSC: 23641 | *y[1] w[*]; P{w[+mC]=UASp YFP. Rab7}21/SM5* |
| Genetic reagent (*D. melanogaster*) | *UAS-Cp1*$^{EP}$ | Bloomington *Drosophila* Stock Center | BDSC: 15957 | *y[1] w[67c23]; P{w[+mC] y[+mDint2]=EPgy2}Cp1 [EY05806]* |
| Genetic reagent (*D. melanogaster*) | *UAS-Vha100-1*$^{EP}$ | Bloomington *Drosophila* Stock Center | BDSC: 63269 | *w[1118]; P{w[+mC]=EP}Vha100-1 [G4514]/TM6C, Sb[1]* |
| Genetic reagent (*D. melanogaster*) | *UAS-Trpml* | Kartik Venkatachalam | Flybase: FBti0162438 | |
| Genetic reagent (*D. melanogaster*) | *UAS-Vha44*$^{EP}$ | Bloomington *Drosophila* Stock Center | BDSC: 20140 | *y[1] w[67c23]; P{w[+mC] y[+mDint2] =EPgy2}Vha44[EY02202]* |
| Genetic reagent (*D. melanogaster*) | *UAS-VhaSFD*$^{EP}$ | Bloomington *Drosophila* Stock Center | BDSC: 15758 | *y[1] w[67c23]; P{w[+mC] y[+mDint2]=EPgy2}VhaSFD [EY04644]/CyO* |
| Genetic reagent (*D. melanogaster*) | *UAS-Rab7*$^{DN}$ | Bloomington *Drosophila* Stock Center | BDSC: 9778 | *y[1] w[*]; P{w[+mC]=UASp YFP. Rab7.T22N}06* |
| Genetic reagent (*D. melanogaster*) | *UAS-Cp1*$^{RNAi}$ | Bloomington *Drosophila* Stock Center | BDSC: 32932 | *y[1] sc[*] v[1] sev[21]; P{y[+t7.7] v[+t1.8]=TRiP. HMS00725}attP2* |
| Genetic reagent (*D. melanogaster*) | *UAS-Vha100-1*$^{RNAi}$ | Bloomington *Drosophila* Stock Center | BDSC: 26290 | *y[1] v[1]; P{y[+t7.7] v[+t1.8]=TRiP.JF02059}attP2* |
| Genetic reagent (*D. melanogaster*) | *UAS-Trpml*$^{RNAi}$ | Bloomington *Drosophila* Stock Center | BDSC: 31294 | *y[1] v[1]; P{y[+t7.7] v[+t1.8]=TRiP.JF01239}attP2* |
| Genetic reagent (*D. melanogaster*) | *UAS-Vha44*$^{RNAi}$ | Bloomington *Drosophila* Stock Center | BDSC: 33884 | *y[1] sc[*] v[1] sev[21]; P{y[+t7.7] v[+t1.8]=TRiP.HMS00821}attP2* |

*Continued on next page*

*Appendix 1—key resources table continued*

| Reagent type (species) or resource | Designation | Source or reference | Identifiers | Additional information |
|---|---|---|---|---|
| Genetic reagent (*D. melanogaster*) | UAS-VhaSFD<sup>RNAi</sup> | Bloomington *Drosophila* Stock Center | BDSC: 40896 | *y[1] sc[*] v[1] sev[21]; P{y[+t7.7] v[+t1.8]=TRiP.HMS02144}attP40* |
| Genetic reagent (*D. melanogaster*) | UAS-Atg6<sup>RNAi</sup> | Bloomington *Drosophila* Stock Center | BDSC: 35741 | *y[1] sc[*] v[1]; P{y[+t7.7] v[+t1.8]=TRiP.HMS01483}attP2* |
| Genetic reagent (*D. melanogaster*) | UAS-Atg18a<sup>RNAi</sup> | Bloomington *Drosophila* Stock Center | BDSC: 34714 | *y[1] sc[*] v[1]; P{y[+t7.7] v[+t1.8]=TRiP.HMS01193}attP2* |
| Genetic reagent (*D. melanogaster*) | UAS-Atg1 | Bloomington *Drosophila* Stock Center | BDSC: 51655 | *y[1] w[*]; P{w[+mC]=UAS-Atg1.S}6B* |
| Genetic reagent (*D. melanogaster*) | UAS-Atg7 | Bloomington *Drosophila* Stock Center | NA | *w[1118]; P{w[+mC]=UAS-Atg7}* |
| Genetic reagent (*D. melanogaster*) | UAS-Atg101<sup>RNAi</sup> | Bloomington *Drosophila* Stock Center | BDSC: 34360 | *y[1] sc[*] v[1]; P{y[+t7.7] v[+t1.8]=TRiP.HMS01349}attP2* |
| Genetic reagent (*D. melanogaster*) | UAS-Atg8a<sup>RNAi</sup> | Bloomington *Drosophila* Stock Center | BDSC: 34340 | *y[1] sc[*] v[1]; P{y[+t7.7] v[+t1.8]=TRiP.HMS01328}attP2* |
| Genetic reagent (*D. melanogaster*) | UAS-Atg5<sup>RNAi</sup> | Bloomington *Drosophila* Stock Center | BDSC: 27551 | *y[1] v[1]; P{y[+t7.7] v[+t1.8]=TRiP.JF02703}attP2* |
| Genetic reagent (*D. melanogaster*) | UAS-Atg5<sup>RNAi</sup> | Bloomington *Drosophila* Stock Center | BDSC: 34899 | *y[1] sc[*] v[1]; P{y[+t7.7] v[+t1.8]=TRiP.HMS01244}attP2* |
| Genetic reagent (*D. melanogaster*) | UAS-Atg8a<sup>RNAi</sup> | Bloomington *Drosophila* Stock Center | BDSC: 28989 | *y[1] v[1]; P{y[+t7.7] v[+t1.8]=TRiP.JF02895}attP2 e[*]/TM3, Sb[1]* |
| Genetic reagent (*D. melanogaster*) | UAS-Atg14<sup>RNAi</sup> | Bloomington *Drosophila* Stock Center | BDSC: 55398 | *y[1] v[1]; P{y[+t7.7] v[+t1.8]=TRiP.HMC04086}attP2* |
| Genetic reagent (*D. melanogaster*) | UAS-Atg16<sup>RNAi</sup> | Bloomington *Drosophila* Stock Center | BDSC: 34358 | *y[1] sc[*] v[1] sev[21]; P{y[+t7.7] v[+t1.8]=TRiP.HMS01347}attP2* |
| Genetic reagent (*D. melanogaster*) | UAS-Atg16<sup>RNAi</sup> | Bloomington *Drosophila* Stock Center | BDSC: 58244 | *y[1] v[1]; P{y[+t7.7] v[+t1.8]=TRiP.HMJ22265}att P40/CyO* |
| Genetic reagent (*D. melanogaster*) | UAS-Atg17<sup>RNAi</sup> | Bloomington *Drosophila* Stock Center | BDSC: 36918 | *y[1] sc[*] v[1]; P{y[+t7.7] v[+t1.8]=TRiP.HMS01611}attP2/ TM3, Sb[1]* |

*Appendix 1—key resources table continued*

| Reagent type (species) or resource | Designation | Source or reference | Identifiers | Additional information |
|---|---|---|---|---|
| Genetic reagent (*D. melanogaster*) | *UAS-Atg6[RNAi]* | Bloomington *Drosophila* Stock Center | BDSC: 28060 | *y[1] v[1]; P{y[+t7.7] v[+t1.8]=TRiP.JF02897}attP2* |
| Genetic reagent (*D. melanogaster*) | *UAS-Atg8b[RNAi]* | Bloomington *Drosophila* Stock Center | BDSC: 34900 | *y[1] sc[*] v[1]; P{y[+t7.7] v[+t1.8]=TRiP.HMS01245}attP2* |
| Genetic reagent (*D. melanogaster*) | *UAS-bchs[RNAi]* | Bloomington *Drosophila* Stock Center | BDSC: 42517 | *y[1] v[1]; P{y[+t7.7] v[+t1.8]=TRiP.HMJ02083}attP40* |
| Genetic reagent (*D. melanogaster*) | *UAS-Atg8b[RNAi]* | Bloomington *Drosophila* Stock Center | BDSC: 27554 | *y[1] v[1]; P{y[+t7.7] v[+t1.8]=TRiP.JF02706}attP2* |
| Genetic reagent (*D. melanogaster*) | *UAS-Atg9[RNAi]* | Bloomington *Drosophila* Stock Center | BDSC: 28055 | *y[1] v[1]; P{y[+t7.7] v[+t1.8]=TRiP.JF02891}attP2* |
| Genetic reagent (*D. melanogaster*) | *UAS-Atg18a[RNAi]* | Bloomington *Drosophila* Stock Center | BDSC: 28061 | *y[1] v[1]; P{y[+t7.7] v[+t1.8]=TRiP.JF02898}attP2* |
| Genetic reagent (*D. melanogaster*) | *UAS-Gyf[RNAi]* | Bloomington *Drosophila* Stock Center | BDSC: 28896 | *y[1] v[1]; P{y[+t7.7] v[+t1.8]=TRiP.HM05106}attP2* |
| Genetic reagent (*D. melanogaster*) | *Atg6[00096]* | Bloomington *Drosophila* Stock Center | BDSC: 11487 | *ry[506] P{ry[+t7.2]=PZ}Atg6 [00096]/TM3, ry[RK] Sb[1] Ser[1]* |
| Genetic reagent (*D. melanogaster*) | *UAS-Atg4b[RNAi]* | Bloomington *Drosophila* Stock Center | BDSC: 56046 | *y[1] v[1]; P{y[+t7.7] v[+t1.8]=TRiP.HMS04249}attP2* |
| Genetic reagent (*D. melanogaster*) | *UAS-Atg17[EP]* | Bloomington *Drosophila* Stock Center | BDSC: 15618 | *y[1] w[67c23]; P{w[+mC] y[+mDint2]=EPgy2}Atg 17[EY03045]* |
| Genetic reagent (*D. melanogaster*) | *bchs[58]* | Bloomington *Drosophila* Stock Center | BDSC: 9887 | *y[1] w[*]; P{w[+mC]=EP} EP2299, bchs[58]/CyO* |
| Genetic reagent (*D. melanogaster*) | *UAS-Atg4a[RNAi]* | Bloomington *Drosophila* Stock Center | BDSC: 35740 | *y[1] sc[*] v[1]; P{y[+t7.7] v[+t1.8]=TRiP.HMS01482}attP2* |
| Genetic reagent (*D. melanogaster*) | *UAS-Atg4a[RNAi]* | Bloomington *Drosophila* Stock Center | BDSC: 44421 | *y[1] v[1]; P{y[+t7.7] v[+t1.8]=TRiP.GLC01355}attP40* |
| Genetic reagent (*D. melanogaster*) | *UAS-Atg10[RNAi]* | Bloomington *Drosophila* Stock Center | BDSC: 40859 | *y[1] v[1]; P{y[+t7.7] v[+t1.8]=TRiP.HMS02026}attP40* |

*Continued on next page*

*Appendix 1—key resources table continued*

| Reagent type (species) or resource | Designation | Source or reference | Identifiers | Additional information |
|---|---|---|---|---|
| Genetic reagent (*D. melanogaster*) | UAS-Atg16$^{RNAi}$ | Bloomington *Drosophila* Stock Center | BDSC: 34358 | y[1] sc[*] v[1]; P{y[+t7.7] v[+t1.8]=TRiP.HMS01347}attP2 |
| Genetic reagent (*D. melanogaster*) | UAS-Atg9$^{RNAi}$ | Bloomington *Drosophila* Stock Center | BDSC: 34901 | y[1] sc[*] v[1]; P{y[+t7.7] v[+t1.8]=TRiP.HMS01246}attP2 |
| Genetic reagent (*D. melanogaster*) | UAS-lt$^{RNAi}$ | Bloomington *Drosophila* Stock Center | BDSC: 34871 | y[1] sc[*] v[1]; P{y[+t7.7] v[+t1.8]=TRiP.HMS00190}attP2 /TM3, Sb[1] |
| Genetic reagent (*D. melanogaster*) | UAS-Atg7$^{RNAi}$ | Bloomington *Drosophila* Stock Center | BDSC: 34369 | y[1] sc[*] v[1]; P{y[+t7.7] v[+t1.8]=TRiP.HMS01358}attP2/ TM3, Sb[1] |
| Genetic reagent (*D. melanogaster*) | Atg7$^{d06996}$ | Bloomington *Drosophila* Stock Center | BDSC: 19257 | w[1118]; P{w[+mC]=XP} Atg7[d06996]/CyO |
| Genetic reagent (*D. melanogaster*) | UAS-Atg4a$^{RNAi}$ | Bloomington *Drosophila* Stock Center | BDSC: 28367 | y[1] v[1]; P{y[+t7.7] v[+t1.8]=TRiP.JF03003}attP2 |
| genetic reagent (*D. melanogaster*) | UAS-Atg8a$^{RNAi}$ | Bloomington *Drosophila* Stock Center | BDSC: 58309 | y[1] v[1]; P{y[+t7.7] v[+t1.8]=TRiP.HMJ22416}attP40 |
| Genetic reagent (*D. melanogaster*) | Bchs$^{17}$ | Bloomington *Drosophila* Stock Center | BDSC: 9888 | y[1] w[*]; P{w[+mC]=EP} EP2299, bchs[17]/CyO |
| Genetic reagent (*D. melanogaster*) | UAS-Atg8a$^{EP}$ | Bloomington *Drosophila* Stock Center | BDSC: 10107 | w[1118] P{w[+mC]=EP} Atg8a[EP362] |
| Genetic reagent (*D. melanogaster*) | UAS-Atg2$^{EP}$ | Bloomington *Drosophila* Stock Center | BDSC: 17156 | w[1118]; P{w[+mC]=EP}Atg2[EP3697]/ TM6B, Tb[1] |
| Genetic reagent (*D. melanogaster*) | UAS-Atg7$^{RNAi}$ | Bloomington *Drosophila* Stock Center | BDSC: 34369 | y[1] sc[*] v[1]; P{y[+t7.7] v[+t1.8]=TRiP.HMS01358}attP2 /TM3, Sb[1] |
| Genetic reagent (*D. melanogaster*) | UAS-Atg13$^{RNAi}$ | Bloomington *Drosophila* Stock Center | BDSC: 40861 | y[1] v[1]; P{y[+t7.7] v[+t1.8]=TRiP.HMS02028}attP40 |
| Genetic reagent (*D. melanogaster*) | UAS-Atg14$^{RNAi}$ | Bloomington *Drosophila* Stock Center | BDSC: 40858 | y[1] v[1]; P{y[+t7.7] v[+t1.8]=TRiP.HMS02025}att P40/CyO |
| Genetic reagent (*D. melanogaster*) | UAS-Atg3$^{EP}$ | Bloomington *Drosophila* Stock Center | BDSC: 16429 | y[1] w[67c23]; P{w[+mC] y[+mDint2]=EPgy2}Atg3 [EY08396] |

*Continued on next page*

*Appendix 1—key resources table continued*

| Reagent type (species) or resource | Designation | Source or reference | Identifiers | Additional information |
|---|---|---|---|---|
| Genetic reagent (*D. melanogaster*) | UAS-Atg18b[RNAi] | Bloomington *Drosophila* Stock Center | BDSC: 34715 | *y[1] sc[*] v[1]; P{y[+t7.7] v[+t1.8]=TRiP.HMS01194}attP2* |
| Genetic reagent (*D. melanogaster*) | UAS-Atg2[RNAi] | Bloomington *Drosophila* Stock Center | BDSC: 27706 | *y[1] v[1]; P{y[+t7.7] v[+t1.8]=TRiP.JF02786}attP2* |
| Genetic reagent (*D. melanogaster*) | Snap29[B6-21] | Bloomington *Drosophila* Stock Center | BDSC: 56818 | *w[*]; P{ry[+t7.2]=neoFRT}42D Snap29[B6-21]/CyO, P{w[+mC]=GAL4 twi.G}2.2, P{w[+mC]=UAS-2xEGFP}AH2.2* |
| Genetic reagent (*D. melanogaster*) | UAS-Atg3[RNAi] | Bloomington *Drosophila* Stock Center | BDSC: 34359 | *y[1] sc[*] v[1]; P{y[+t7.7] v[+t1.8]=TRiP.HMS01348}attP2* |
| Genetic reagent (*D. melanogaster*) | UAS-Atg2[RNAi] | Bloomington *Drosophila* Stock Center | BDSC: 34719 | *y[1] sc[*] v[1]; P{y[+t7.7] v[+t1.8]=TRiP.HMS01198}attP2* |
| Genetic reagent (*D. melanogaster*) | UAS-Atg18b[RNAi] | Bloomington *Drosophila* Stock Center | BDSC: 34715 | *y[1] sc[*] v[1]; P{y[+t7.7] v[+t1.8]=TRiP.HMS01194}attP2* |
| Genetic reagent (*D. melanogaster*) | Atg4b[P0997] | Bloomington *Drosophila* Stock Center | BDSC: 36340 | *y[1] w[*]; P{w[+mC]=lacW}Atg4b[P0997]* |
| Genetic reagent (*D. melanogaster*) | UAS-bchs-HA | Bloomington *Drosophila* Stock Center | BDSC: 51636 | *y[1] w[*]; P{w[+mC]=UAS bchs.HA}32* |
| Cell line (*Homo sapiens*) | HeLa stably expressing TFEB:GFP | Shawn Ferguson (*Roczniak-Ferguson et al., 2012*) | | |
| Biological sample (*Homo sapiens*) | Control (non-neurological) and ALS postmortem motor cortex tissue | Ravitz laboratory (UCSD) through Target ALS Consortium; Brain Resource Center at JHMI | | |
| Antibody | Rabbit polyclonal anti-dsRed | Clontech | Cat#63249, RRID:AB_10013483 | 1:1000 for IF |
| Antibody | Mouse monoclonal anti- poly-ubiquitin | Enzo Life Sciences | Cat#BML-PW8805, RRID:AB_10541434 | 1:200 for IF |
| Antibody | Rabbit polyclonal anti-ref(2)P | Gabor Juhasz laboratory (*Pircs et al., 2012*) | | 1:1000 for IF; 1:1000 for WB |
| Antibody | Guinea pig polyclonal anti-Mitf | Francesca Pignoni laboratory (*Zhang et al., 2015b*) | | 1:500 for IF |
| Antibody | Rat monoclonal anti-HA | Roche | Cat# 11867423001, RRID:AB_390918 | 1:200 for IF |

*Continued on next page*

*Appendix 1—key resources table continued*

| Reagent type (species) or resource | Designation | Source or reference | Identifiers | Additional information |
|---|---|---|---|---|
| Antibody | Chicken polyclonal anti-GFP | Abcam | Cat# ab13970, RRID:AB_300798 | 1:1000 for IF; 1:1000 for WB |
| Antibody | Guinea pig polyclonal anti-Cp1 | Patrick Dolph laboratory (*Kinser and Dolph, 2012*) | | 1:2500 for WB |
| Antibody | Mouse monoclonal anti- beta actin (clone C4) | EMD Millipore | Cat# MAB1501, RRID:AB_2223041 | 1:1000 for WB |
| Antibody | Rabbit polyclonal anti- TFEB | Bethyl Biosciences | Cat# A303-673A, RRID:AB_11204751 | 1:2000 for WB |
| Antibody | Rabbit polyclonal anti-Histone H3 | Cell Signaling | Cat# 9715, RRID:AB_331563 | 1:1000 for WB |
| Antibody | Mouse monoclonal anti-FLAG | Sigma Aldrich | Cat# F3165, RRID:AB_259529 | 1:1000 for IF |
| Recombinant DNA reagent | pSF-CAG-Amp | Oxford Genetics | Cat# 0G504 | |
| Sequence-based reagent | *Actin* forward | Integrated DNA Technologies | q-RT-PCR primer | 5'- GCGCGGTTACTCTTTCACCA-3' |
| Sequence-based reagent | *Actin* reverse | Integrated DNA Technologies | q-RT-PCR primer | 5'- ATGTCACGGACGATTTCACG-3' |
| Sequence-based reagent | $G_4C_2$ repeats forward (*UAS-30R*) | Integrated DNA Technologies | q-RT-PCR primer | 5'-GGGATCTAGCCACCATGGAG-3' |
| Sequence-based reagent | $G_4C_2$ repeats reverse (*UAS-30R*) | Integrated DNA Technologies | q-RT-PCR primer | 5'-TACCGTCGACTGCAGAGATTC-3' |
| Sequence-based reagent | *Mitf* forward | Integrated DNA Technologies | q-RT-PCR primer | 5'-AGTATCGGAGTAGATGTGCCAC-3' |
| Sequence-based reagent | *Mitf* reverse | Integrated DNA Technologies | q-RT-PCR primer | 5'-CGCTGAGATATTGCCTCACTTG-3' |
| Sequence-based reagent | *Vha16-1* forward | Integrated DNA Technologies | q-RT-PCR primer | 5'- TCTATGGCCCCTTCTTCGGA-3' |
| Sequence-based reagent | *Vha16-1* reverse | Integrated DNA Technologies | q-RT-PCR primer | 5'- AATGGCAATGATACCCGCCA-3' |
| Sequence-based reagent | *Vha68-2* forward | Integrated DNA Technologies | q-RT-PCR primer | 5'-CAAATATGGACGTGTCTTCGCT-3' |
| Sequence-based reagent | *Vha68-2* reverse | Integrated DNA Technologies | q-RT-PCR primer | 5'- CCGGATCTCCGACAGTTACG-3' |
| Sequence-based reagent | *Vha55* forward | Integrated DNA Technologies | q-RT-PCR primer | 5'- CGGGACTTTATCTCCCAGCC-3' |
| Sequence-based reagent | *Vha55* reverse | Integrated DNA Technologies | q-RT-PCR primer | 5'-TGACCTCATCGAGAATGACCAG-3' |
| Sequence-based reagent | *Vha44* forward | Integrated DNA Technologies | q-RT-PCR primer | 5'-TGGACTCGGAGTACCTGACC-3' |

*Continued on next page*

*Appendix 1—key resources table continued*

| Reagent type (species) or resource | Designation | Source or reference | Identifiers | Additional information |
|---|---|---|---|---|
| Sequence-based reagent | *Vha44* reverse | Integrated DNA Technologies | q-RT-PCR primer | 5'-CGTCACGTTGAACAGGCAGTA-3' |
| Sequence-based reagent | *Vha100-2* forward | Integrated DNA Technologies | q-RT-PCR primer | 5'-TGTTCCGTAGTGAGGAGATGG-3' |
| Sequence-based reagent | *Vha100-2* reverse | Integrated DNA Technologies | q-RT-PCR primer | 5'-TCACGTTCACATTCAAGTCGC-3' |
| Sequence-based reagent | *Atg8a* forward | Integrated DNA Technologies | q-RT-PCR primer | 5'-GGTCAGTTCTACTTCCTCATTCG-3' |
| Sequence-based reagent | *Atg8a* reverse | Integrated DNA Technologies | q-RT-PCR primer | 5'-GATGTTCCTGGTACAGGGAGC-3' |
| Sequence-based reagent | *Atg9* forward | Integrated DNA Technologies | q-RT-PCR primer | 5'- ACACGCCTCGAAACAGTGG-3' |
| Sequence-based reagent | *Atg9* reverse | Integrated DNA Technologies | q-RT-PCR primer | 5'-TCAAGGTCCTCGATGTGGTTC-3' |
| Sequence-based reagent | *ref(2)P forward* | Integrated DNA Technologies | q-RT-PCR primer | 5' - ATGCCGGAGAAGCTGTTGAA - 3' |
| Sequence-based reagent | *ref(2)P reverse* | Integrated DNA Technologies | q-RT-PCR primer | 5' - ATCAGCGTCGATCCAGAAGG - 3' |
| Commercial assay or kit | SuperScript III First-Strand Synthesis System | Thermo Fischer Scientific | Cat #18080051 | |
| Commercial assay or kit | NE-PER Nuclear and Cytoplasmic Extraction Kit | Thermo Fischer Scientific | Cat #78833 | |
| Commercial assay or kit | BCA Assay | Thermo Fischer Scientific | Cat #23227 | |
| Commercial assay or kit | 4–15% Mini-PROTEAN TGX Precast Gel | Bio-Rad | Cat #4561083 | |
| Commercial assay or kit | One Shot TOP10 Chemically Competent *E. coli* | Thermo Fischer Scientific | Cat# C404006 | |
| Commercial assay or kit | Faststain | G-Biosciences | Cat #786–34 | |
| Commercial assay or kit | SYBR Select Master Mix | Thermo Fischer Scientific | Cat #4472908 | |
| Chemical compound, drug | Blotting Grade Blocker (nonfat dry milk) | Bio-Rad | Cat #1706404 | |
| Chemical compound, drug | Lipofectamine 2000 | Thermo Fischer Scientific | Cat #11668019 | |

*Appendix 1—key resources table continued*

| Reagent type (species) or resource | Designation | Source or reference | Identifiers | Additional information |
|---|---|---|---|---|
| Chemical compound, drug | Mifepristone (RU486) | Millipore Sigma | Cat #M8046 | |
| Chemical compound, drug | Rapamycin | Selleckchem | Cat #S1039 | |
| Chemical compound, drug | D-(+)-Trehalose dihydrate | Millipore Sigma | Cat #T0167 | |
| Chemical compound, drug | TRIzol | Thermo Fischer Scientific | Cat #15596018 | |
| Chemical compound, drug | Protease Inhibitor Cocktail | Roche | Cat#11873580001 | |
| Software, algorithm | ImageJ | https://imagej.nih.gov/ij/ | | |
| Software, algorithm | GraphPad Prism 8 | https://www.graphpad.com/scientific-software/prism/ | | |
| Software, algorithm | IMARIS 9 | https://imaris.oxinst.com/ | | |
| Software, algorithm | Adobe Illustrator CC 2018 | https://www.adobe.com/products/illustrator | | |
| Software, algorithm | Image Pro Insight 9.1 | http://www.mediacy.com/imagepro | | |
| Software, algorithm | WinWCP | https://pureportal.strath.ac.uk/en/datasets/strathclyde-electrophysiology-software-winwcp-winedr | | |

