## [Decision Letter]

**Acceptance summary:**

In this study investigators use a logical and well-designed experimental course from identification of Ref(2)p/p62 as a modifier of neurodegeneration in *Drosophila* and its accumulation as a driver of degeneration to altered nuclear transport of the autophagic transcription factor Mitf as the basis for disruption of autophagy in this model of C9. This is a timely study, linking to major pathways previously associated with C9 disease-nuclear/cytoplasmic transport and autophagy. Importantly, key steps in the pathway identified in *Drosophila* are confirmed in human cell models and patient material.

**Decision letter after peer review:**

Thank you for submitting your article "TFEB/Mitf links impaired nuclear import to autophagolysosomal dysfunction in C9-ALS" for consideration by *eLife*. Your article has been reviewed by three peer reviewers, one of whom is a member of our Board of Reviewing Editors, and the evaluation has been overseen by K VijayRaghavan as the Senior Editor. The following individual involved in review of your submission has agreed to reveal their identity: Udai Pandey (Reviewer #2).

The reviewers have discussed the reviews with one another and the Reviewing Editor has drafted this decision to help you prepare a revised submission.

Summary:

This study reports that alteration in the nuclear transport of transcription factor Mitf (homolog of TFEB) results in a decrease in transcription of genes encoding components of the autophagic pathway thereby disrupting autophagy and proteostasis. Overall, this is a well thought-out and timely study that links two important physiological processes in the context of ALS/FTD. However, addition of analysis/data that further support linking Mitf TFEB mislocalization to disease pathogenesis in C9orf72-mediated ALS would enhance the impact of this work.

Essential revisions:

1) In Figure 1: The authors used only one G4C2 repeat line in this figure. Testing additional G4C2 repeat expansion lines (36 and 58 repeats) to determine if knocking down ref(2)p modulates their phenotypes.

2) Measure nuclear Mitf levels (not N/C ratio) in Figure 3C-D, Figure 5B and Figure 7B.

3) Examine endogenous TFEB distribution in Figure 7.

4) Determine if Mitf expression rescues GR30 or other DPR toxicity.

5) Examine measures of autophagy flux in Figure 6—figure supplement 1.

6) Determine if emb knockdown reverses p62 accumulation in 30R-expressing *Drosophila*.

---

## [Author Response]

Essential revisions:1) In Figure 1: The authors used only one G4C2 repeat line in this figure. Testing additional G4C2 repeat expansion lines (36 and 58 repeats) to determine if knocking down ref(2)p modulates their phenotypes.

We thank the reviewers for this suggestion. We have now tested knock-down of *ref(2)P* in multiple G4C2 repeat expansion lines (30, 36, and 44 repeats) and show this data in a revised Figure 1. Importantly, two independent *ref(2)P* RNAi lines significantly suppress 36R-mediated eye degeneration (Figure 1C-D), and the eclosion phenotype caused by 44R expression in motor neurons (vGlut-GAL4) is strongly suppressed by *ref(2)P* knockdown and enhanced by *ref(2)P* overexpression (Figure 1F). These data are similar to what we previously showed for the 30R lines (Figure 1A-B and E), and confirm that *ref(2)P* is a potent suppressor of G4C2-repeat mediated phenotypes.

2) Measure nuclear Mitf levels (not N/C ratio) in Figure 3C-D, Figure 5B and Figure 7B.

In the nuclear transport field, it is customary to display nucleocytoplasmic transport as the ratio of nuclear to cytoplasmic protein levels. However, we agree that since we are specifically interested in nuclear Mitf signaling in our study, showing Nuclear Mitf levels may be more intuitive to readers, and we now show the quantitation of Nuclear Mitf or TFEB levels in all of our figures (Figure 4B, D; Figure 5B; and Figure 7B). Given the variability from cell-to-cell, nuclear Mitf and TFEB levels are best represented as a percent of total rather than absolute levels. Importantly, with these new analyses, our conclusions are unchanged, namely that Nuclear Mitf is reduced by 30R expression (Figure 4A-D) and is restored by inhibiting nuclear export with *emb* knockdown (Figure 5), and also that nuclear TFEB fails to increase in response to starvation in HeLa cells expressing 47R.

3) Examine endogenous TFEB distribution in Figure 7.

We have attempted to examine endogenous TFEB distribution in the HeLa cells shown in Figure 7 using multiple commercially available human TFEB antibodies that have been reported to work in human cells using Immunohistochemistry. Unfortunately, we do not observe depletion of TFEB with siRNA-mediated knockdown (Author response image 1), nor do we observe nuclear translocation by treating cells with the mTOR inhibitor Torin1 (Li et al., 2018; Napolitano et al. (2018) Nat Commun9, 3312, Author response image 2), strongly suggesting that the immunostaining we observe with these antibodies is nonspecific. We have also consulted experts in this field who confirm that available antibodies against vertebrate TFEB do not work well in immunohistochemistry (Drs. Connie Cortes and Albert La Spada, personal communications). To circumvent this problem, the Ferguson lab generated a stably transfected cell line that expresses TFEB:GFP at low levels, and they have previously demonstrated that TFEB:GFP translocates to the nucleus under starvation conditions (Roczniak-Ferguson et al., 2012). Thus, the TFEB:GFP stable cell line is the best tool available to monitor TFEB localization in human cells.

**Author response image 1. respfig1:** Multiple commercially available TFEB antibodies fail to show specific staining as demonstrated by lack of reduced staining with TFEB siRNA. HEK293T cells stained with (A) Abcam anti-TFEB ab174745, (B) Bethyl anti-TFEB A303-673A and (C) Proteintech 13372-1-AP in control (left) or transfected with TFEB siRNA (ONTarget-Plus, Dharmacon) (right).

**Author response image 2. respfig2:** Multiple commercially available TFEB antibodies fail to show specific staining as demonstrated by lack of increase in nuclear localization of TFEB after treatment with Torin1, an mTOR inhibitor known to induce nuclear translocation of TFEB. HEK293T cells stained with (A) Abcam anti-TFEB ab174745, (B) Bethyl anti-TFEB A303-673A and (C) Proteintech 13372-1-AP in control (left) or treated with 2μm Torin for 2 hours.

Furthermore, as an alternative approach, we have performed nuclear-cytoplasmic fractionation in post-mortem motor cortex (Figure 7C-D) to determine the relative levels of endogenous TFEB in the nucleus and cytoplasm. We believe this approach is the best method to confirm that TFEB is downregulated and mislocalized in C9-ALS brain.

4) Determine if Mitf expression rescues GR30 or other DPR toxicity.

We have now performed these experiments in the *Drosophila* eye. As shown in Figure 6—figure supplement 1, Mitf duplication does not rescue the severe rough eye phenotype observed with overexpression of GR36. This suggests that Mitf rescues phenotypes induced by the G4C2 repeat rather than polyGR, suggesting that loss of Mitf function does not contribute to polyGR-mediated toxicity. Indeed, this is consistent with our observation that overexpressing GR50-mCherry does not alter the ability of starvation to induce nuclear translocation of TFEB in HeLa cells (Figure 7—figure supplement 1).

5) Examine measures of autophagy flux in Figure 6—figure supplement 1.

We thank the reviewers for this suggestion. We have now examined multiple measurements of autophagic flux in 30R-expressing flies coexpressing the Mitf Dp (duplication) line to stimulate lysosomal function, similar to the genetic manipulations that we previously performed in Figure 6—figure supplement 1 to modulate lysosome function downstream of Mitf. Importantly, the Mitf duplication significantly suppresses both the expanded lysosome phenotype (Figure 6E-F) and the p62 accumulation phenotype (Figure 6G-H) in 30R-expressing flies, suggesting an improvement in autophagic flux. Together, these new data support our model that increasing Mitf expression and target genes suppresses autophagolysosomal defects in C9-ALS/FTD.

6) Determine if emb knockdown reverses p62 accumulation in 30R-expressing *Drosophila.*

We have now performed this experiment in motor neurons (Figure 5G-H), and interestingly find that emb knockdown does not rescue p62:GFP accumulation. We suspect that the reason for the lack of rescue in this case is that inhibiting nuclear export with emb knockdown causes slight p62 accumulation on its own (Figure 5G), consistent with our observations that either knockdown or overexpression of RanGAP causes p62:GFP accumulation (Figure 5—figure supplement 1). In the Discussion, we state: “Interestingly, genetic inhibition of nuclear export or increase in Mitf expression are able to strongly rescue autophagosome and lysosome phenotypes and neurodegeneration, but do not result in complete clearance of p62 accumulations (Figures 5-6). Additional studies will be needed to better understand the relationship between p62 aggregation, autophagy, nucleocytoplasmic transport, and neurodegeneration.”